

# The chemical characteristics of rainwater and wet atmospheric deposition fluxes at two urban sites and one rural site in Côte d' Ivoire.

**Mohamed L. Kassamba-Diaby [1], Corinne Galy-Lacaux [2], Veronique Yoboué [1], Jonathan E. Hickman [3], Kerneels Jaars [4], Sylvain Gnamien [1], Richmond Konan [1], Eric Gardrat [2], Siele Silué [5]**

[1] Laboratoire des Sciences de la Matière, de l'environnement et de l'énergie Solaire / Université Félix Houphouët-Boigny, Abidjan BPV 34, Côte d'Ivoire

[2] Laboratoire d'Aérologie, Université Toulouse III Paul Sabatier / CNRS, Toulouse, France

[3] Center for Climate Systems Research, Columbia University, New York, USA

[4] Atmospheric Chemistry Research Group, Research Focus Area: Chemical Resource Beneficiation, North-West University, Potchefstroom, South Africa

[5] University Peloforo Gbon Coulibaly/ Korhogo, Côte d'Ivoire

Corresponding author: Mohamed Lamine Kassamba Diaby, email: diabykassamba@yahoo.fr, Université Felix Houphouet Boigny, address: 01 BP V 34, 01 Abidjan

Key words

Rainwater, chemical composition, wet deposition, acidification, neutralization, eutrophication

urban and rural sites, Cote d'Ivoire, Africa

Abstract

In this study, we characterized the chemical composition of precipitation at three sites in Côte d'Ivoire representative of a south-north transect. Two urban sites have been selected in the framework of the Pollution and Health in Urban Areas (PASMU) project: one located in Abidjan in the coastal climate zone and the other located in Korhogo in the northern climate zone. The third site is the International Network to study Deposition and Atmospheric chemistry in Africa (INDAAF) rural site of Lamto representative of a wet savanna and located in the central climate zone. This work documents a three-year time period (2018-2020) with 221 samples, 239 samples and 143 samples which have been collected in Abidjan, Lamto and Korhogo, respectively. Annual and monthly VWM concentration of major ions ($Na^+$, $K^+$, $Mg^{2+}$, $Ca^{2+}$, $Cl^-$, $NO_3^-$ $SO_4^{2-}$, $NH_4^+$, $HCOO^-$, $CH_3COO^-$, $C_2H_5COO^-$, $C_2O_4^{2-}$) in rainwater have been calculated and were found to follow the following patterns: $Ca^{2+}$> $Cl^-$> $Na^+$> $NH_4$> $SO_4^{2-}$> Tcarb > $NO_3^-$> $Mg^{2+}$> $HCOO^-$ > $CH_3COO^-$ > $K^+$ > $H^+$> $C_2O_4^{2-}$> $C_2H_5COO^-$ in Abidjan, $NH_4^+$ > $HCOO^-$ > $Ca^{2+}$ > $NO_3^-$> $CH_3COO^-$ >$H^+$ > $Cl^-$ >$Na^+$>





$SO_4^{2-} > Mg^{2+} > K^+ > Tcarb > C_2O_4^{2-} > C_2H_5COO^-$ in Lamto and $Ca^{2+} > NH_4^+ > Na^+ > HCOO^- > NO_3^- > Cl^-$
$> K^+ > CH_3COO^- > SO_4^{2-} > H^+ > Mg^{2+} > Tcarb > C_2O_4^{2-} > C_2H_5COO^-$ in Korhogo. The average pH values
are respectively 5.76, 5.31, 5.57 for Abidjan, Lamto and Korhogo with a preponderance of mineral acidity
at the urban sites representing respectively 69 % and 52% of the total acidity contribution in Abidjan and
Korhogo while the organic acidity is more important in Lamto representing 62 % of the total acidity
contribution. Dry seasons contribute to 46%, 74 % and 86% to the total measured chemical content of
precipitation respectively at Abidjan, Lamto and Korhogo. During dry seasons, Lamto and Korhogo rainfalls
are more impacted by biomass burning source and continental air mass loaded in terrigenous compounds
than Abidjan. Conversely, during wet seasons Abidjan rainfalls are more impacted by oceanic air mass from
guinean gulf rich in sea salt than Lamto and Korhogo.
1.Introduction
Atmospheric deposition represents a key mechanism in anthropogenic impacts on the environment.
Atmospheric deposition includes wet and dry processes, and is the major removal pathway of atmospheric
pollutants and thus contributes to the equilibrium of the earth-atmosphere biogeochemical balance (Vet et
al., 2014; Laouali et al., 2021; Fu et al., 2021; Galy-Lacaux et al., 2009). Limiting anthropogenic impacts
on atmospheric deposition is considered fundamental for addressing several sustainable development goals
such food security, climate change, human health and biodiversity (Rockström et al., 2009; Fowler et al.,
2013; Fu et al., 2021). The study of deposition processes and the determination of deposition fluxes is
important for understanding the spatial and temporal evolution of the chemical composition of the
atmosphere and of the biogeochemical cycles of elements such as nitrogen, carbon and sulfur. Where
biogeochemical cycles are strongly affected by anthropogenic activities, atmospheric deposition can act as
a source of nutrients but also as a source of toxins (Bobbink et al., 2010; Zhang et al., 2007a; Whelpdale et
al., 1997).
Wet deposition plays a key role in removing both gaseous and particulate pollutants from the atmosphere
and thus influences atmospheric chemistry (Seinfeld and Pandis, 1998; Laouali et al., 2012). Rain chemical
composition provides insights into the evolution of the chemical composition of the atmosphere, and is
influenced by numerous factors including the type and strength of natural/anthropogenic sources of
atmospheric compounds, long-range transport,  the origin of continental air masses, as well as removal
processes related to the intensity and temporal distribution pattern of rainfall  (Vet et al., 2014; Akpo et al.,
2015; Keresztesi et al., 2019). In addition, rainfall composition is useful for understanding direct impacts
on ecosystems and is an important indicator in the determination of the pollution levels in urban areas
(Moreda-Piñeiro et al. 2014; Martins et al. 2019; Gao et al. 2020; Günzel 2020). Lack of accurate
descriptions of deposition processes and thorough evaluation with high-quality measurements remain a
major weakness of global deposition modelling.  This is particularly true in tropical regions, which are often



affected by convective rainfall regime, and where long-term high-quality data on deposition are scarce
(Fowler et al., 2013; Vet et al., 2014; Fu et al., 2021).
To date, the most recent study in this context is the global assessment of precipitation chemistry and
deposition carried out under the auspices of the World Meteorological Organization (WMO) -Global
Atmospheric Watch (GAW) Scientific Advisory Group Total Atmospheric Deposition (SAG TAD), which
aims to characterize precipitation chemical composition and to quantify deposition fluxes (wet, dry, total)
of sulfur, nitrogen, acidity, sea salt, organic acids and phosphorus at global and continental scales. The study
compares two temporal reference periods: 2000-2002 and -2005-2007 (Vet et al., 2014). The conclusion of
that assessment led to some recommendations to address major gaps and uncertainties in global ion
concentration and deposition measurements. One of these recommendations emphasizes the lack of
measurements in tropical regions and the weakness of the spatial coverage in different continents such as
South America, parts of India and Africa (Vet et al., 2014; Fu et al., 2021).
The assessment recognized the importance of the unique long-term quality-controlled database in Africa
provided by the International Network to study Deposition and Atmospheric chemistry in Africa (INDAAF,
https://indaaf.obs-mip.fr) even though the number of measurements stations remain low. The INDAAF
program, initiated in 1994, aims to study atmospheric composition and wet and dry deposition fluxes in
Africa. It is part of the Deposition of Biogeochemically important Trace Species (DEBITS) activity of the
International Global atmospheric Chemistry (IGAC) as well as an official contributor network to the
GAW/WMO program and a labeled component of the Aerosol Cloud and Trace gases Research
Infrastructure (ACTRIS).
The INDAAF activity is based on a regional long term monitoring network of 10 stations representative of
three major African ecosystems covering dry savanna, wet savanna and equatorial forest (https://indaaf.obs-
mip.fr (Laouali et al., 2021). High quality measurements of atmospheric chemistry (rainwater, aerosol and
gaseous chemical composition) are performed on a multi-year scale. Many synthesis studies which are
representative of ecosystem rural sites, or which rely on the comparison of the eco-systemic transect dry
savannas, wet savannas-forests, have been published (Laouali et al., 2012; Yoboué et al., 2005a; Akpo et
al., 2015; Galy-Lacaux et al., 2009; Galy-Lacaux and Modi, 1998). Although these works have characterized
precipitation chemistry and deposition in Africa representative of rural areas, there are no studies to our
knowledge that consider urban areas in Africa. In the context of the rapid urbanization and demographic
explosion in Africa (United Nations, Department of Economic and Social Affairs, Population Division
2017; Kaba et al. 2020), it is important to improve our understanding of  urbans areas precipitation
composition and the ion deposition fluxes at the seasonal and annual scale in order to assess the evolution
of atmospheric composition and the potential impacts of pollutants under the influence of increasing human
activity in cities and megacities in developing countries.





In Côte d'Ivoire, the percentage of the national population living in urban areas is expected to increase to
60% by 2025 and exceed 70% by 2050 (UN World Urban Population , 2011). Increased urbanization and
population will likely be accompanied by increasing pollutants emissions from fossil fuel consumption, as
in other countries such as in China which experienced extreme pollution levels following its fast economic
development (Wang et al., 2008; Zhang et al., 2007).
The present study proposes to investigate precipitation chemistry and wet deposition fluxes over a south
north transect in Côte d' Ivoire, considering three measurements sites, including two urban and one rural
site. This work is performed in collaboration with two major monitoring programs: the Air Pollution and
Health in Urban Areas (PASMU) implemented in 2018 to study the atmospheric chemical pollution and
impacts on human health in the economic capital of Côte d'Ivoire (Abidjan) and in the regional city of
Korhogo in relation to meteorological parameters and emission sources (Gnamien et al., 2021); and the
INDAAF program, which includes the site of Lamto, representative of a soudano-guinean wet savanna.
The main objective of the present study is to establish the characteristics of the chemical composition of
precipitation and the deposition fluxes of two urban areas and one rural area in Côte d' Ivoire, together
representative of a continental south-north transect. The goals of this study are: (1) to document over a three-
year time period (2018-2020) the rainwater chemical composition and the deposition fluxes of soluble ions,
including concentrations of major ions, the variation of pH, concentrations of sea salts, the neutralizing
capacity of precipitation, and ion enrichment factors, (2) to provide a better understanding of ion sources
and the climatology that influence annual, seasonal and monthly precipitation content and (3) to analyze the
intra-annual and seasonal variability of precipitation composition and associated wet deposition fluxes for
the different ionic species. This study offers a baseline record for urban sites in African cities against which
future changes in emissions and potential environmental impacts can be evaluated, and responds to
international recommendations that emphasize the scarcity of deposition measurements on the African
continent, recognized at a global scale to be a continent faced with major environmental sustainability issues
(World Bank, 2017; World Meteorological Organization, 2021).
2. Materials and Methods
2.1. Sites description



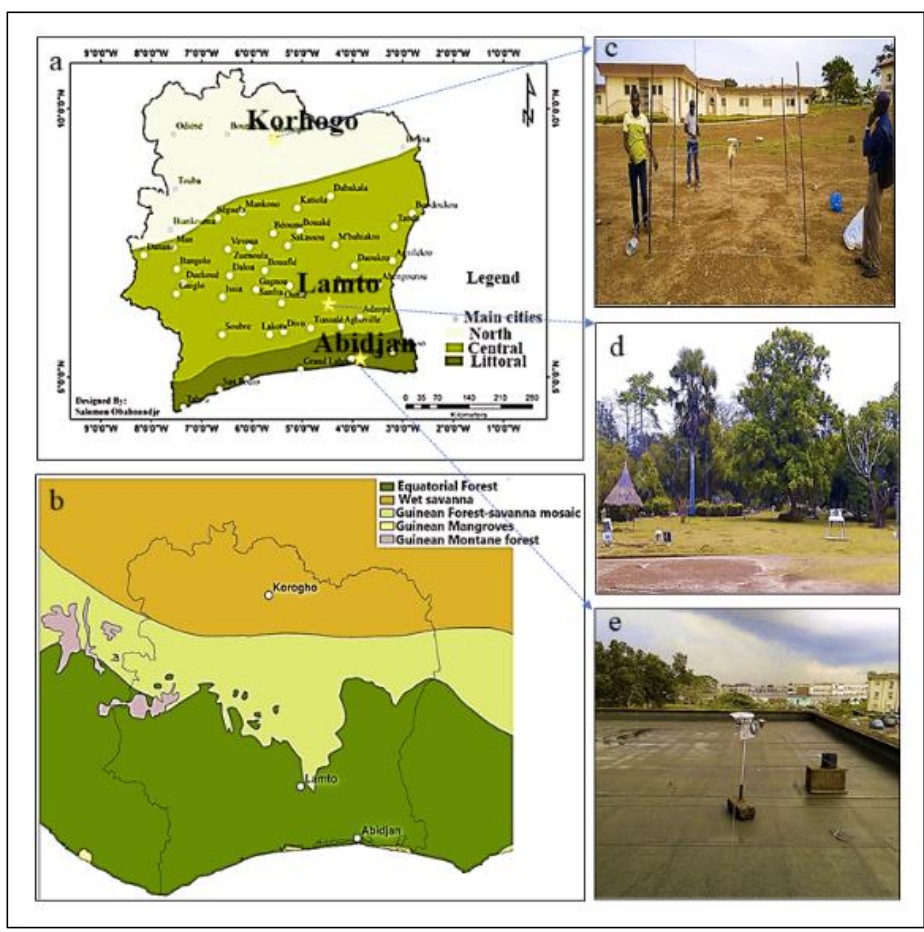

Figure 1: Locations of the three measurement sites on the Abidjan-Lamto-Korhogo transect: (a): map of climatic subdivision in Cote d' Ivoire adapted from (Kouadio et al, 2007); (b): map of ecoregions subdivision in Côte d'Ivoire (c): of Korhogo; (d): Lamto; (e): Abidjan.

This study considers three measurement sites, two urban and one rural, located along a south-north transect in Côte d'Ivoire (Figure 1). The two urban sites, Abidjan and Korhogo, have been selected and studied in the framework of the PASMU program, and are respectively located in the south and north of Côte d'Ivoire. The modes of transportation, the types of fuel used by households and the population density make it possible to distinguish and characterize both of these urban sites. It is worth noting that in Côte d'Ivoire, southern cities are generally more populated and industrialized than those in the north. For example, Abidjan's population is 10 times larger than that of Korhogo (Gnamien et al., 2021) and according to (Fall et al., 2016), Ivorian cities can be divided into three categories : global connectors, which are cities such as Abidjan, that generate the economies of urbanization necessary for innovation, increasing returns to scale activities and global competitiveness; regional connectors, that are cities such as Korhogo, which generate



the local economies necessary for efficient regional trade and transportation, and local connectors, that are
cities that generate the economies of scale necessary to release agricultural potentials of their regions.
The first urban site is located in Abidjan (5° 20' 43." N; 4° 1' 27." W) which is a metropolitan area on the
south-east coast of Côte d'Ivoire and considered to be the economic capital of the country, Abidjan is the
largest city in Côte d' Ivoire with a population over 4 707 404 million, which is approximately 20 % of the
population in Côte d'Ivoire, and a surface area of 2119 km$^2$ (INS, 2014). This city is an autonomous district
divided in 13 suburbs. The sampling site was on the roof top of the Institut de Recherche et de Development
(IRD) building, which is in the suburb of Cocody, in the vicinity of the University Felix Houphouet Boigny.
The major pollution sources in the city are fossil fuel consumption from traffic of motorized vehicles,
household coal burning and emissions from industrials activities. (Yao et al., 2016) estimated that the
national fleet of vehicles was 636 551 in 2016 with 80% in Abidjan (498.531 vehicles).
The second urban site is located in the city of Korhogo (9° 28' N; 5° 36' 51" W), which is situated in the
north of Côte d'Ivoire, approximatively 635 km from Abidjan in the savannas district. Korhogo is spread
over an area of 12.50 km$^2$ and has a population of 243 048 inhabitants, according to the last population
census in 2014 (INS, 2014). Korhogo is strongly influenced by agricultural activities, even though it is an
urban area. According to Bassett et al. (2018), Korhogo is the epicenter of the cotton and cashew boom
culture, which is dependent on fertilizers and pesticides for crop production. In terms of the level of
urbanization and industrialization, Korhogo is far from the level of megacities such as Abidjan, although it
has recorded substantial population growth since the political crisis in 2002, resulting in the surface of the
city increasing from 3300 ha in 2000 ha to 10000 ha in 2019 (Sangare et al., 2021). Nonetheless, it has a
relatively low level of urbanization and industrialization compared to Abidjan. The mode of transport is
dominated by two-wheeled vehicles. This trend is also observed at the level of public transport with the
emergence of motorcycle cabs, Taxi-motos", which constitute one of the principal means of transport since
the prohibition of four-wheeled taxi vehicles at the time of the political-military crisis in September 2002
(Roger et al., 2016).
The third sampling site, Lamto, represents a super-site of the INDAAF project and is located in the central
part of the country, at the tip of the "V Baoule". Lamto (6°13' N, 5°02' W) is located in the region of Agnéby-
Tiassa, in the department of Tiassalé, about 165 km in the north-west of Abidjan and 433 km in the south-
east of Korhogo. It is in a natural reserve that covers approximately 2600 ha and is representative of a
soudano-guinean wet savanna with the so-called, gallery forest along the Bandama river (Gautier et
al.,1990).
2.2. Climatology
West African climate depends on the position of the Intertropical zone of convergence (ITZC), which is the
limit between a cool and humid marine air mass (Monsoon) and a warm and dry Saharan air mass





(Harmattan). Climate is largely is influenced by the ITZC variability at regional scale in Côte d' Ivoire.
Indeed, the extreme latitudinal positions of ITCZ zone in January (5◦N) and in August (22◦N) divide the
climate into three distinct zones: the Northern climatic zone, the Central climatic zone and the Coastal
Climatic zone (Kouadio et al., 2003) (Figure 1-a) . The three experimental sites are each situated in a
different zone: Abidjan in the Coastal Climatic zone, Lamto in the central climatic zone and Korhogo in the
Northern climatic zone. In These climatic zones , we have different pluviometric regimes: the tropical
regime of transition, the humid tropical regime of transition, the dimmed equatorial regime of transition, the
equatorial regime of transition and littoral equatorial regime of transition (Goula et al., 2010).
Meteorological parameters from 2018 to 2020 (monthly temperature and relative humidity) were provided
by the SODEXAM (Society of exploitation and airport development, Aeronautics and Meteorology) for
Abidjan and Korhogo, as well as long-term rainfall databases for the periods 1980-2020 and 1990-2020
respectively. From 2018 to 2020 in Abidjan, we used rainfall measured by the EVIDENCE project (Extreme
rainfall events, vulnerability to flooding and water contamination) that installed tipping bucket rain gauge
(tilting of the bucket for 0.5mm) and Précis Mécanique® (rain interception cone 1.5 m from the ground and
with an area equal to 400 cm$^2$). Tipping bucket dates (day month year hour minute second) are recorded in
a HOBO Pendant® UA-003-64 data logger. The rain gauge data were collected monthly. During each visit,
the devices were cleaned and the tipping volume of the buckets was checked in the urban site of Abidjan.
At the Lamto site, the long-term monitoring program (INDAAF) provides air temperature, humidity and
rainfall data for the period (Diawara et al., 2014).




















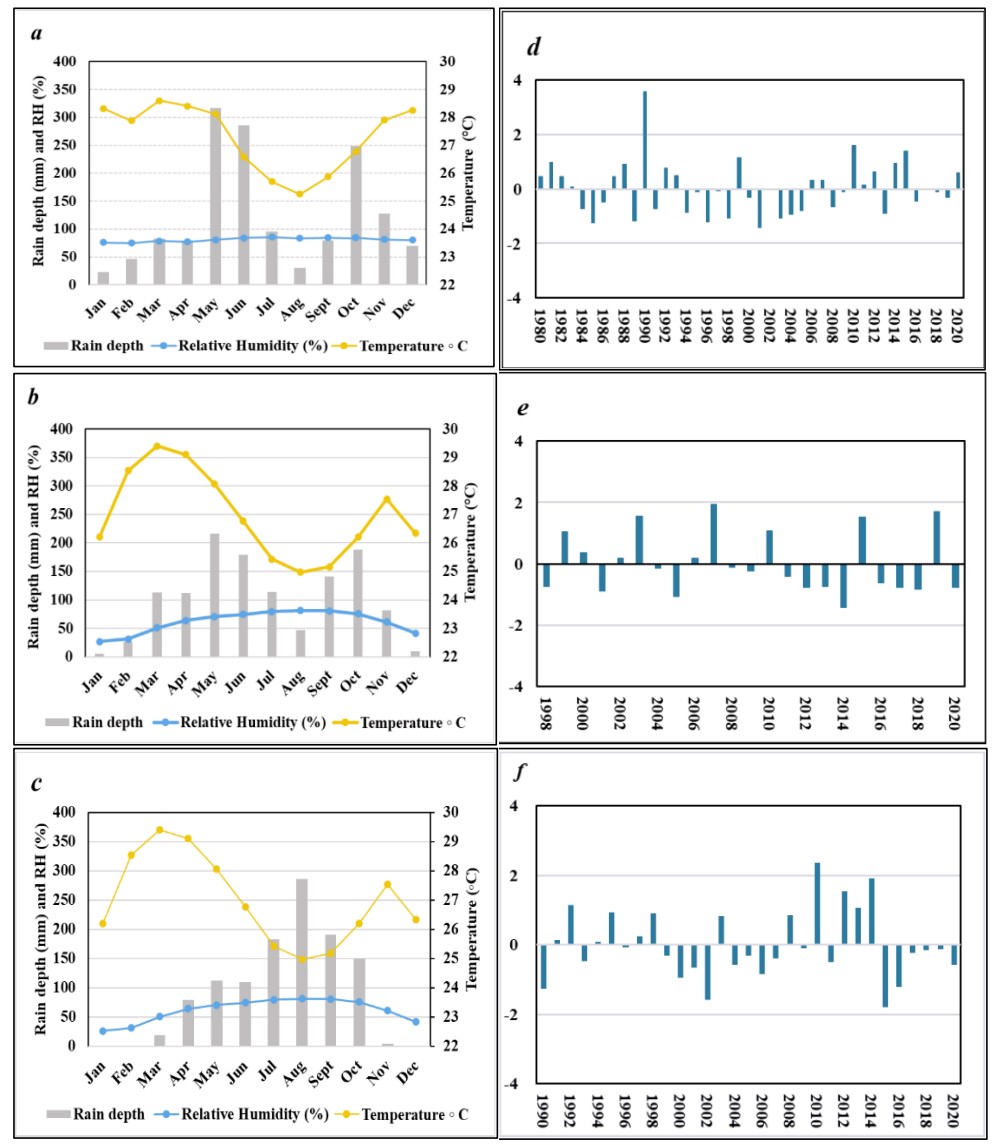

Figure 2: Monthly mean meteorological parameters measured at Abidjan (a), Lamto (b), and Korhogo (c) in 2019 (Air temperature (°C), Relative Humidity (%), Rain depth (mm)); Annual Inter variability Index (AII) for Abidjan (1980-2020) (d), Lamto (1998-2020) (e) and Korhogo (1990-2020) (f).



Abidjan is characterized by a bimodal rainfall regime defined by two wet seasons and two dry seasons. A
long-wet season lasts from March to July and a short-wet season from October to November. A long dry
season lasts from December to February and a short dry season from August to September (Leroux et al.
2001). Abidjan belongs to the coastal climatic zone characterized by a first rainfall maximum in June and a
second in September (Figure 2). The annual rainfall is in the range 784-3388.9 mm with an annual mean of
1522 ± 518 mm for the period 1980-2020. Lamto is situated in the central climatic zone, providing the site
with a mild climate warm and wet. We also distinguish two wet seasons and two dry seasons: one long wet
season that extends from March to July, mainly influenced by the monsoon air masses; one long dry season
from December to February, influenced by the Saharan air masses (harmattan) (Diawara et al., 2014); one
short dry season limited to the month of August; and one short wet season from September to November.
In terms of rainfall features, Lamto belongs to the equatorial coastal transition regime which has an annual
rainfall ranging from 991.9 to 1548.5 mm, with a mean annual rainfall of 1229 ± 165 mm for the period

233    1998-2020.

The Korhogo site is part of the north climatic zone, which is characterized by a unimodal rainfall regime,
varying between a single wet season from April to October and a single dry season from November to
March. Korhogo belongs to the tropical regime of transition, characterized by a maximum of rainfall in
August. The annual rainfall is between 867-1612 mm, with an annual mean of 1187 ± 179 mm from 1990
to 2020.
Observations in Abidjan show a weak fluctuation of air temperature and relative humidity during the studied
period (2018-2020). The annual mean temperature and relative humidity are approximately 27.30 °C ± 1.10
and 80 % ± 3.89 respectively (Figure 2). The maximum temperature is observed in March (28.65±1.85 °C)
and minimum in August (25.3 ± 0.20°C). Lamto meteorological parameters show a similar profile to
Abidjan. The maximum temperature is observed in February (30.83 ± 0.25 °C), followed by a gradual drop
until August, when the minimum temperature is observed (26°C ± 0.21) (Figure 2). The mean relative
humidity over the study period is 77% ± 5.53. Air temperature and rainfall evolve in the opposite way.
During the gradual drop in temperature at the beginning of the year, we observe a gradual increase in
monthly rainfall (Figure 2). Korhogo presents a mean annual temperature and relative humidity of 27.00 ±
0.08°C and 60% ± 0.81 respectively. Temperature increases from January to April to reach a monthly
maximum in April (29.1 ± 0.26 °C). The temperature then decreases gradually to a minimum in August
(25.2 ± 0.30 °C) that coincides with the beginning of the rainy season. The temperature then rises again until
november, followed by a sharp decrease in December.
Figure 2  presents also the calculation of the Annual Inter variability Index (AII) (Sarr, 2009), which enables
the characterization of the general patterns of precipitation over the study period. According to Sarr, the
degree of drought is a function of the index (AII) of precipitation. If AII > 2, the humidity is extreme, and





AII < − 2 represents extreme drought. The intermediate values of the index are classified as follows: 1 < AII < 2, 0 < AII <1, − 1 < AII < 0 and − 2 < AII < − 1, corresponding to high humidity, moderate humidity, moderate drought and strong drought respectively. Considering data in Abidjan from 1980 to 2020, we have 20 years of surplus and 18 years of deficit compared to an average rainfall of approximately 1522 mm, with the maximum rainfall recorded in 1990 (3338.9 mm) and the minimum in 2001 (784 mm). According to the classification of Sarr, 2020 (AII = + 0.14) can be considered as a moderately wet period while the other two years 2018 (AII = -0.08), 2019 (AII =-0.31) can be classified as moderately dry years. For Lamto, from 1998 to 2020, there are 8 years of surplus and 13 years of deficit compared to an average rainfall of approximately 1229 mm. The maximum rainfall was recorded in 2007 (1548.5 mm) and the minimum in 2014 (991.9 mm). For the three years of the study period, 2019 (AII =+1.7) can be considered as a strongly wet period while 2018 (AII =-0.8) and 2020 (AII =-0.8) can be classified as moderately dry years. Finally, at Korhogo from 1990 to 2020, we observe 12 years of surplus and 18 years of deficit with the respect to the average rainfall of 1187 mm. The maximum occurred in 2010 (1612 mm) and the minimum in 2015 (866.7 mm). The three years of the study period, 2018-2020, are all years of deficit, with AII index values of -0.14, -0.15, -0.58 respectively. According to Sarr classification, they can be considered as moderately dry periods.

2.3. Sample collection

| Sites | ABIDJAN | | | LAMTO | | | KORHOGO | | |
|---|---|---|---|---|---|---|---|---|---|
| Year | 2018 | 2019 | 2020 | 2018 | 2019 | 2020 | 2018 | 2019 | 2020 |
| Pt (mm) | 1477.7 | 1355.4 | 1593.7 | 1090.9 | 1508.2 | 1101.4 | 1160.7 | 1162.5 | 1083 |
| Inter annual Variability (%) | -2.91 | -10.94 | 4.71 | -0.80 | 22.71 | -10.38 | -2.21 | -2.06 | -8.76 |
| Pc (mm) Nc | 825 (56) | 1006.20 (81) | 1288.30 (84) | 1077.60 (91) | 1459.40 (70) | 988 (78) | 745.55 (48) | 862.80 (52) | 783.10 (43) |
| % TP (%) | 56 | 74 | 81 | 99 | 97 | 90 | 64 | 74 | 72 |
| % PCL Annually % (quarterly) | 75 (0,1,1,1) | 100 (1,1,1,1) | 100 (1,1,1,1) | 100 (1,1,1,1) | 100 (1,1,1,1) | 100 (1,1,1,1) | 75 (0,1,1,1) | 100 (1,1,1,1) | 100 (1,1,1,1) |

Table 1: Rainwater collection at Abidjan, Lamto and Korhogo (2018-2020): Annual Total Precipitation (Pt, mm), Interannual variability (%), Collected precipitation (Pc, mm) and Number of collected rain events (Nc), Percent total precipitation (%TP), Annual percent coverage length (%PCL) and in brackets: %PCL for each quarter (0 and 1 means 0% and 100% respectively).





Precipitation sampling at the three sites was performed using a semi-automatic collector of precipitation
designed for the INDAAF (International Network to study Deposition and Atmospheric composition in
Africa) network (http://indaaf.obs-mip.fr). The equipment characteristics as well as the sampling protocols
have been fully described in several studies (Galy-Lacaux et al., 2009; Laouali et al., 2012; Akpo et al.,
2015). In brief, an automatic precipitation sampler made of up a single-use polyethylene bag, avoiding
aerosol deposit before the onset of the rain and a precipitation sensor automatically controls the aperture of
the sampler cover, which hermetically closes the polyethylene bag. At each site, a local technician collected
each rain event in a 50 mL Greiner tube, that was immediately placed in an on-site freezer (-18°C). The rain
sampling protocol follows the WMO/GAW international standards recommendations (WMO, 2004). After
collection, samples were sent for analysis at the Laboratoire d'Aerologie (Laero) in Toulouse, observing
very strict temperature regulation during the voyage. Table 1 presents the annual total precipitation (Pt) in
mm, the percent total precipitation (%TP) and the interannual variability as a percentage relative to the mean
annual rainfall for the 1980-2020 period, 1998-2020 period and 1990 -2020 period respectively for Abidjan,
Lamto and Korhogo. As defined by (WMO, 2004), (%TP) is the ratio between the annual precipitation (Pt)
and the collected precipitation (Pc). The annual and quarterly Percent Coverage Length (%PCL) which is
the percent of the summary period (e.g., month, season, year) for which information is available on whether
precipitation occurred or not is also indicated in Table 1. From April 2018 to December 2020 in Abidjan,
the total rainfall amount was 4426.8 mm and the collected rain samples represent a total of 3119.50 mm
with 221 samples. For Lamto, from January 2018 to December 2020 the total rainfall amount was 3700.5
mm and the collected rain samples represent a total rainfall amount of 3525 mm with 239 samples. For
Korhogo, from May 2018 to December 2020, the total rainfall was 3406.2 mm and the collected rain samples
represent a total of 2391.45 mm with 143 samples.
The %TP shows that rainwater collection in 2018 was not a good indicator of actual rainfall in Abidjan and
Korhogo, with values of 56 % and 64 % respectively. Lamto has good %TP values for all years. However,
for comparison purposes, only 2019 and 2020 will be considered for computing the annual volume weighed
mean (VWM) and Wet deposition fluxes (WD), and the average of the two years 2019-2020, where the
mean collection rate is respectively 78 % ,94 %, 73 % and the PCL is 100% in Abidjan, Lamto and Korhogo.
Data from January 2018 to December 2020 will be used to calculate monthly Volume Weighed Mean
(VWM) and Wet Deposition (WD) according to the quarterly %PCL. In reference to the WMO international
standards, we assume the precipitation collection at Abidjan, Lamto and Korhogo in 2019 and 2020 can be
considered as representative of the studied period according to the parameters calculated in Table 1.







2.4. Analytical procedures and quality assurance / quality control

Major inorganic (Na$^+$, K$^+$, Mg$^{2+}$, Ca$^{2+}$, Cl$^-$, NO$_3^-$ SO$_4^{2-}$, NH$_4^+$) and organic ( HCOO$^-$, CH$_3$COO$^-$, C$_2$H$_5$ COO$^-$, C$_2$O$_4^{2-}$) ions were determined by Ionic Chromatography (IC) at Laero as described in (Galy-Lacaux and Modi, 1998; Akpo et al., 2015). The Ionic chromatographic analysis is performed using a Thermo ICS5000+ and an ICS 1100 Ionic chromatographs with two automated samplers (AS50). The eluents for anions and cations are NaOH and MSA, respectively. Certified ionic standards are used for IC calibration. pH is measured with an ATI Orion 350 instrument with a combined electrode (ATI Orion model 9252) filled with KCl (4 M) and saturated with AgCl. Two standard solutions (WTW) at pH 4.01 and 7.00 are used for its calibration. The precision is 0.01pH unit.

Since 1996, the Laboratoire d'Aerologie has participated to the bi-annual inter-laboratory comparison study (LIS) of WMO-GAW precipitation quality assurance program. Results are available under the reference 700106 at the following address: http://qasac-americas.org/. According to these WMO inter-comparison tests, analytical precision is 5% or better for all ions, within the uncertainties on all measured ionic values presented here. Data quality is further ensured by calculating the ion difference for each sample to consider the ionic balance (WMO, 2004). Analyses were performed on 221 rain samples collected in Abidjan, 239 rains samples collected in Lamto and 143 rains samples collected in Korhogo (Table 1). Results indicate in Abidjan, Lamto and Korhogo respectively 208 or 94 % of collected samples, 236 or 98 % of collected samples and 127 or 89 % are in the WMO acceptance range and will be considered in all the calculations presented in the result sections.

2.5 Satellite data

We used version 1.6 of the CrIS-Fast Physical Retrieval (CFPR)-NH$_3$ product ( Shephard et al. 2015; Mark W. Shephard et al. 2020). CFPR is a physical retrieval of an atmospheric profile of NH$_3$ derived from minimizing residuals between the measured and simulated spectra. The product compares well with in situ and ground-based FTIR observations (Dammers et al., 2017; Kharol et al., 2018). The CrIS sensor has an NH$_3$ detection limit of ~0.3 to 0.9 ppb (Shephard et al., 2020), which varies depending on the atmospheric conditions. Comparing the NASA/NOAA SNPP satellite radiances from both CrIS and VIIRS instruments show that the stability of CrIS is very good. The VIIRS – CrIS daily mean brightness temperature difference shows a trend of -0.40 ± 0.03 mK per year at the wavelength of 10.76 um for a ~273 K average scene (David Tobin, personal communication). The most recent update of the CFPR (v1.6) identifies and accounts for non-detect values below the sensor's detection limit, which reduces the previously reported small positive bias for the lower range of total column concentrations ($<5\text{x}10^{15}$ molecules cm$^{-2}$). For this study, only



daytime observations from 2018 to 2020 are used. Note that observations for April-July 2019 are not
available, due to instrument error.
We used Level 3 (L3) data at 0.25°×0.25°resolution from the NASA tropospheric NO2 standard product
from the Ozone Monitoring Instrument (OMI) aboard NASA's Aura satellite. OMI is a nadir-viewing
spectrometer in sun-synchronous orbit with near-daily global coverage that measures solar backscatter in
the UV-visible range (Krotkov et al., 2017). The OMI product relies on air mass factors calculated with the
assistance of an atmospheric chemical transport model, and is sensitive to model representations of
emission, chemistry and transport data. These are generally poorly constrained for regions not commonly
analyzed in chemical transport models such as sub-Saharan Africa region (McLinden et al., 2014). The L3
product includes only pixels that are at least 70% cloud-free, which may introduce additional bias: since the
product relies on nearly cloud-free retrievals, greater sunlight may induce higher photochemical rates. The
Level 2 OMI-NO$_2$ product has shown good agreement with in situ and surface-based observations(Lamsal
et al., 2014). For our analyses of satellite retrievals over Abidjan, Lamto and Korhogo, we selected
observations centered around the 0.25° grid cell containing each site to create a 1° field of NO$_2$ or NH$_3$
VCDs. The mean of this 1° grid cell was used as an estimate of VCDs over the site.
2.6. Calculations and statistics
The monthly Volume Weighed Mean (VWM) concentrations as well as the annual VWM concentrations in
µeq.L$^{-1}$ for each ion were calculated using methods described by (Laouali et al., 2012; Conradie et al., 2016)

370    :

$$VWM = \frac{\sum_{i=1}^{N} C_i \cdot P_i}{\sum_{i=1}^{N} P_i} \quad (1)$$

Where Ci in µeq. L$^{-1}$ is the concentration of a given chemical element for each rain event, Pi is rainfall depth
for each rain event in mm. N is the number of rain events.
The annual as well as monthly wet deposition fluxes for all ionic species is expressed in kg. ha$^{-1}$. yr$^{-1}$ and
calculated by these following formulae (Laouali et al., 2021):
$$WD = (VWM / c_i) * M_i * P_t / 100000 \quad (2)$$

Where VWM is the concentration in µeq. L$^{-1}$, $c_i$ is the ionic charge, Mi in g.mol$^{-1}$ is the molar mass of each
species and $P_t$ in mm is the annual rain depth for annual wet deposition fluxes and monthly rain depth for
monthly wet deposition fluxes.
The H$^+$ concentrations were calculated from measured pH values: $10^{-pH}$ (3)



Sea Salt Fraction (SSF) to ionic concentrations in rainwater and corresponding enrichment factors (EF) were
calculated according to the method suggested by many authors (Chao and Wong, 2002; Keene and
Galloway, 1986).
$\quad EF_{marine} = [X/Na^+]_{rain} / [X/Na^+]_{sea}$ (4)
$\quad EF_{crustal} = [X/Ca^{2+}]_{rain} / [X/Ca^{2+}]_{crustal}$ (5)
Where X is the concentration of the ion of interest , $Na^+$ is used as the element of reference for marine source
(Kulshrestha et al., 2003) and $Ca^{2+}$ is selected as reference element for continental origin (Safai et al., 2004).
$\quad SSF(X) = (X/[Na+])_{sea} * [Na+]_{rain}$ (6)
$\quad NSS(X) = [X]_{rain} - SSF(X)$ (7)
Where SSF(X) is the marine part of the chemical element X in $\mu eq.L^{-1}$, $[Na^+]_{rain}$ is the concentration of $Na^+$
in rain ($\mu eq\ L^{-1}$) and $[X]/[Na]_{sea}$ is the ratio of species X to $Na^+$ in seawater (Keene et al., 1986). NSS (X)
is the non-marine part of the chemical element X in $\mu eq\ L^{-1}$ and $[X]_{rain}$ is the specific concentration of the
different species X in $\mu eq.\ L^{-1}$, the potential Acidity (pA) is defined as the sum of nitrate, sulfate, formic,
acetic, propionic and oxalic VWM concentrations, supposing that all ions are associated with $H^+$.
$pA=\sum anions=[SO_4^{2-}] +[NO_3^-] +[HCOO^-]+[CH_3COO^-] + [C_2H_5COO^-] +[C_2O_4^{2-}]$ (8)
Fractional Acidity (FA); (Balasubramanian et al., 2001; Cao et al., 2009 ; Lu et al., 2011) is :
$FA = \dfrac{[H^+]}{([NO_3^-] +[SO_4^{2-}]+[HCOO^-]+[CH_3COO^-]+[C_2H_5COO^-]+[C_2O_4^{2-}])}$ (9)
The Neutralization Factor (NF) (Celle-Jeanton et al., 2009; Rastogi and Sarin, 2005) is:
$NF\ xi = \dfrac{[xi]}{([NO_3^-] +[SO_4^{2-}]+[HCOO^-]+[CH_3COO^-]+[C_2H_5COO^-]+[C_2O_4^{2-}])}$ (10)
Where $x_i$ are cations of interest, and all ionic concentrations are expressed in $\mu eq\ L^{-1}$.
The difference between neutralization potential (NP) ($Ca^{2+}+NH_4^+$) and acidic potential (AP) ($SO_4^{2-} + NO_3^-$
) is computing according to the following equation from (Safai et al., 2004): $NP/AP = [Ca^{2+}] +[NH_4^+] /$
$[SO_4^{2-}] + [NO_3^-]$ (11)
For study purposes, we adapted equations 8, 9 ,10 and 11 by integrating the organic acidity which was not
included in the original equations used by the authors in previous studies. This approach enables us to take
into account all the acidity generated on the rainfall sampling sites.




2.7. Back trajectories
In order to determine the impact of air masses on the chemical composition of collected rainwater samples,
The air mass trajectories history for each site for the entire sampling period was determined by calculating
back trajectories with the Hybrid Single-Particle Lagrangian Integrated Trajectory (HYSPLIT) model
(version 4.8), developed by the National Oceanic and Atmospheric Administration (NOAA) Air Resources
Laboratory (ARL) (Draxler and Hess, 1997). The model calculation method is a hybrid between the
Lagrangian approach, using a moving frame of reference for the advection and diffusion calculations as the
trajectories or air parcels move from their initial location, and the Eulerian methodology, which uses a fixed
three-dimensional grid as a frame of reference to compute pollutant air concentrations (The model's name,
no longer meant as an acronym, originally reflected this hybrid computational approach). Hourly-arriving
96-hour back trajectories were calculated at three arrival heights, 100, 1500 and 2500 m respectively, above
ground level. These individual back trajectories were then superimposed, in order to generate monthly or
seasonal overlay back trajectories for the study period, on a frequency map with a 0.2° x 0.2° resolution grid
to show the statistical distribution. The frequency map has a color index that indicates the frequency of the
trajectories passing over the map grid cells. A color scale indicates the number of back trajectories passing
over a grid cell, with dark red indicating the highest percentage of back trajectory overpasses. All calculated
trajectories were visualized using MATLAB R2020b (https://www.mathworks.com/). The trajectories are
constructed using three-dimensional velocity fields, thereby making it an ideal method to incorporate
convective motions and the role thereof in the vertical transport of air masses (L Kok et al., 2021).
3.Results and discussion
3.1. Chemical composition of rainwater and wet deposition fluxes

Annual Volume Weighed Mean (VWM) concentrations and wet deposition fluxes (WD) computed for the
two-year sampling (2019-2020) along the South-North transect Abidjan-Lamto-Korhogo are presented in
the Table A1.  At Abidjan, the chemical signature of the rainwater is characterized by the following ions
concentrations in decreasing order:  $Ca^{2+}> Cl^-> Na^+> NH_4> SO_4^{2-}>$ Tcarb $> NO_3^-> Mg^{2+}> HCOO^- >$
$CH_3COO^- > K^+> H^+> C_2O_4^{2-}> C_2H_5COO^-$. $Ca^{2+}$, $Na^+$, $Cl^-$ and $NH_4^+$ dominate and represent 62 % of the
rainwater total VWM ionic concentrations. At Lamto, the rainwater chemical signature is dominated by the
four following ions: $NH_4^+$, $HCOO^-$, $Ca^{2+}$, and $NO_3^-$, representing 55 % of the total VWM ionic
concentrations. VWM concentrations follow a global pattern in the decreasing concentration order: $NH_4^+ >$
$HCOO^- > Ca^{2+} > NO_3^-> CH_3COO^- >H^+> Cl^- >Na^+> SO_4^{2-} > Mg^{2+} > K^+ >$ Tcarb $> C_2O_4^{2-} > C_2H_5COO^-$.
Korhogo rainwater chemistry composition exhibits a profile dominated by the following ions $Ca^{2+}$, $NH_4^+$,
$Na^+$, and $HCOO^-$, representing 53% of the total VWM ionic concentrations.  The general chemical pattern
in the decreasing concentration order is: $Ca^{2+} > NH_4^+ > Na^+ > HCOO^- > NO_3^- > Cl^- > K^+ > CH_3COO^- >$



$SO_4^{2-} > H^+ > Mg^{2+} > Tcarb > C_2O_4^{2-} > C_2H_5COO^-$. The mean annual ionic load is estimated to 191.20 μeq.
$L^{-1}$, 84.26 μeq. $L^{-1}$, and 111.75 μeq. $L^{-1}$ for Abidjan, Lamto and Korhogo respectively, demonstrating that
the urban precipitations of Abidjan and Korhogo are much more loaded with ions than the rural area of
Lamto.

447        3.1.1. Marine contribution

Sea-salt fractions (SSF), non-sea-salt fractions (NSSF) and enrichment factors (EF) for $K^+$, $Cl^-$, $Mg^{2+}$, $SO_4^{2+}$,
$Ca^{2+}$ ions were calculated according to the methodology outlined in section 2.5 (Table 2).

| Sites | Sea water ratios (Keene et al, 1986) | $Cl^-/Na^+$ | $SO_4^{2-}/Na^+$ | $K^+/Na^+$ | $Ca^{2+}/Na^+$ | $Mg^{2+}/Na^+$ |
|---|---|---|---|---|---|---|
| | | 1.167 | 0.121 | 0.022 | 0.044 | 0.227 |
| ABIDJAN | Ratios in rain | 1.23 | 0.75 | 0.17 | 1.47 | 0.28 |
| | EF$_{MARINE}$ | 1 | 6 | 8 | 33 | 1 |
| | SSF (%) | 94 | 15 | 13 | 3 | 81 |
| LAMTO | Ratios in rain | 1.07 | 0.88 | 0.37 | 1.83 | 0.47 |
| | EF$_{MARINE}$ | 1 | 7 | 17 | 42 | 2 |
| | SSF (%) | 92 | 10 | 5 | 2 | 35 |
| KORHOGO | Ratios in rain | 0.85 | 0.47 | 0.77 | 1.79 | 0.3 |
| | EF$_{MARINE}$ | 1 | 1 | 35 | 41 | 1 |
| | SSF (%) | 100 | 16 | 3 | 2 | 75 |


452            Table 2: Seawater Enrichment Factor (EF) in Abidjan, Lamto and Korhogo

For the three studied sites, the $Cl^-/Na^+$ ratios were close to the sea-salt  ratio of reference (Keene et al., 1986)
and EFs were close to 1, showing that $Cl^-$ is almost 100% marine assuming that most of the sodium is from
a marine source (Cao et al., 2009). $Na^+$ and $Cl^-$  generally originate from sea-salt associated with oceanic
air masses (Niu et al., 2013). $Na^+$ and $Cl^-$ are highly correlated (r=0.94, r=0.82, r=0.83) (Figure 3) in Abidjan,
Lamto and Korhogo respectively and suggest that both ions are mainly of marine origin. The $Mg^{2+}/Na^+$
ratios calculated in Abidjan and Korhogo are close to the seawater reference value and also the EF values
are equal to 1.
This result suggests a marine origin of $Mg^{2+}$ at Abidjan and Korhogo. This result is supported by the strong
correlations calculated between $Mg^{2+}$ and $Na^+$, and $Mg^{2+}$ and $Cl^-$ respectively (r=0.82) and (r=0.94) for
Abidjan, and (r=0.64) and (r=0.62) for Korhogo) (Figure 3).





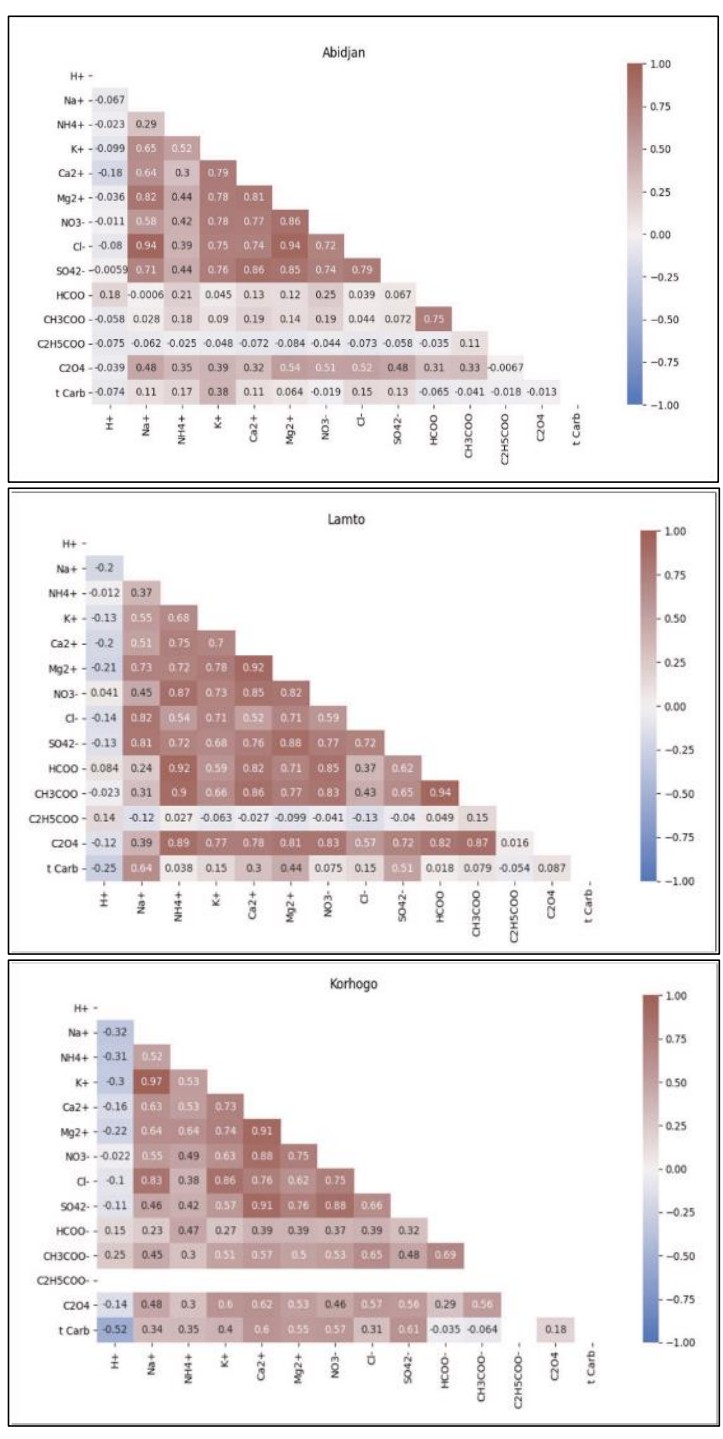

Figure 3: Spearman matrix correlation of rainwater VWM concentrations (µeq. L$^{-1}$) for Abidjan, Lamto and Korhogo.


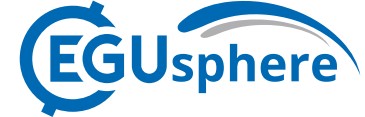

SSF and NSSF calculations indicate that for Abidjan and Korhogo, 81% and 75% of $Mg^{2+}$ is from a marine
origin and that 19% and 25% are from non-marine sources. Lamto presents also a $Mg^{2+}/Na^+$ ratio above the
seawater reference with an EF value close to 2 indicating an additional non-marine contribution. SSF and
NSSF $Mg^{2+}$ fractions are estimated to be 35 % and 65 % respectively. In addition, we found that $Mg^{2+}$ and
$Ca^{2+}$ are highly correlated (r=0.92) (Figure 3) indicating a possible terrigenous origin. The $SO_4^{2-}/Na^+$,
$K^+/Na^+$, $Ca^{2+}/Na^+$ ratios in rainwater at all the sites were found to be higher than the seawater ratios and
corresponding EF values were well above 1.
These high ratios and EF values indicate potential contributions from anthropogenic and crustal sources in
addition to the marine source (Conradie et al., 2016). SSF and NSSF calculations show that $SO_4^{2-}$ at Abidjan,
Lamto and Korhogo is mostly non-marine in origin, with non-marine contributions of 85%, 90% and 84%,
respectively. The marine fraction for $Cl^-$, $SO_4^{2-}$, $K^+$, $Ca^{2+}$, $Mg^{2+}$ were estimated to be approximately 94%,
15%, 13%, 3 % and 81%, respectively at Abidjan, 92%, 10%, 5%, 2 % and 35%, respectively at Lamto, and
100%, 16%, 3%, 2% and 75% respectively at Korhogo. The marine contribution to the total ionic content
for the three sites was computed using the following equation:

485            Marine = $[Na^+]$ + SSF$[Cl^-]$+ SSF$[Mg^{2+}]$ + SSF$[Ca^{2+}]$ +SSF$[K^+]$ +SSF$[SO_4^{2-}]$,

486            where SSF is sea salt fraction


and was estimated to be 65.69 µeq. $L^{-1}$, 11.59 µeq. $L^{-1}$, 25.46 µeq. $L^{-1}$ representing a contribution of 34 %
in Abidjan, 14% in Lamto and 24% in Korhogo (Figure 4). The strong marine contribution in Abidjan is
likely to be related to the coastal location of the city, resulting in a strong influence of monsoon air masses
loaded in sea salt. Similar conclusions are described in several studies (e.g. Hoinaski et al., 2014; Xing et
al., 2017) where high concentrations of  $Na^+$, $Cl^-$ and $Mg^{2+}$ have been attributed to ocean proximity. We
found that the rural site of Lamto records the lowest marine contribution.  To explain these results, air mass
origins influencing Abidjan, Lamto and Korhogo have been studied using back trajectory calculations from
the NOAA HYSPLIT model over the study period 2017-2020 at 1500 m of altitude (Figure 5).















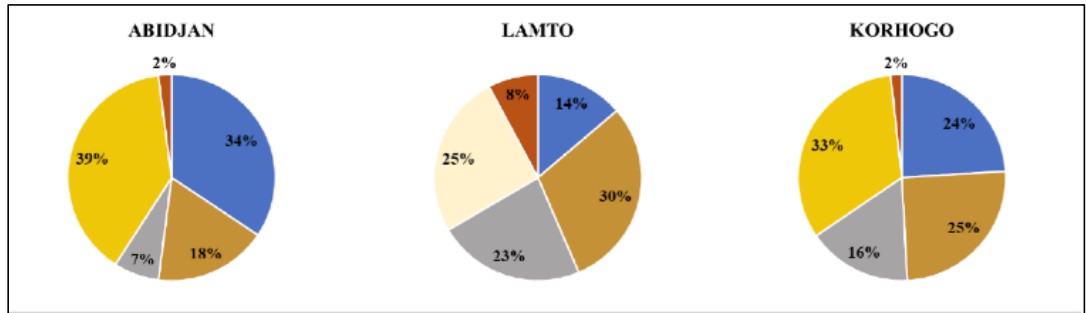

Figure 4: Estimation of marine, nitrogenous, organic, acidity and terrigenous contributions to rain chemical content along the Abidjan-Lamto-Korhogo transect (*terrigenous in this study represents a mixture of terrigenous and anthropogenic sources in urban areas, whereas in the rural Lamto site it represents a mixture of terrigenous and biomass burning sources)



Figure 5: Overlay back-trajectory analyses for air masses arriving at the three sites for the study period (2017-2020). a, d, g: average back-trajectory for the study period for Abidjan, Lamto and Korhogo, respectively. b, e, h: average dry seasons back trajectory for the study period at Abidjan, Lamto and Korhogo respectively. c, f, i: average wet seasons back-trajectory for the study period at Abidjan, Lamto and Korhogo respectively.

Results clearly indicate that the monsoonal oceanic air masses coming from the Guinean Gulf rich in sea-salt aerosols influence the sites of Abidjan, Lamto and Korhogo (Figure 5-c, 5-f, 5-i). This influence is more intense in Abidjan, indicated by a higher percentage of oceanic back trajectories indicated by light blue and yellow cells (Figure 5-c) compared to Lamto with a lower percentage, indicated by a majority dark blue cells (Figure 5-f). Despite its northernmost position and its greater distance from the coast, we found that





Korhogo is more influenced than Lamto by the marine source. Air mass back trajectories show that Korhogo
is influenced both by oceanic air masses coming from the south and from the north-west border, this double
contribution could explain the importance of the marine source contribution in Korhogo rainfall chemical
content (figure 5-i). In West and Central Africa, convective rainfalls generally show a marine signature
related to the boundary layer chemical content and a terrigenous signature from atmospheric levels above
the boundary layer affected by continental air masses. Hot and dry continental air masses originating from
the high-pressure system above the Sahara Desert give rise to dusty Harmattan winds over most of West
Africa from November to February. In summer, moist equatorial air masses originating from the Atlantic
ocean bring annual monsoon rains (Nicholson, 2013). Our results clearly identify these two chemical
signatures during the monsoon season along the studied south-north transect represented by the Abidjan,
Lamto, and Korhogo sites.

556        3.1.2. Terrigenous contribution

$SO_4^{2-}$, $Mg^{2+}$, $K^+$ and $Cl^-$ ratios and crustal enrichments factors (EF) in relation to $Ca^{2+}$ as an element of
reference for crustal materials are presented in Table 3.

| Sites | crustal water ratios (Keene et al 1986) | $Cl^-/Ca^{2+}$ | $SO_4^{2-}/Ca^{2+}$ | $K^+/Ca^{2+}$ | $Mg^{2+}/Ca^{2+}$ |
|---|---|---|---|---|---|
| | | 0.0031 | 0.0188 | 0.504 | 0.561 |
| ABIDJAN | Ratios in rain | 0.84 | 0.65 | 0.12 | 0.19 |
| | $EF_{CRUSTAL}$ | 270 | 27 | 0.23 | 0.3 |
| | NSSF (%) | 6 | 85 | 87 | 19 |
| LAMTO | Ratios in rain | 0.58 | 0.48 | 0.20 | 0.26 |
| | $EF_{CRUSTAL}$ | 26 | 26 | 0.1 | 0.5 |
| | NSSF (%) | 8 | 90 | 96 | 65 |
| KORHOGO | Ratios in rain | 0.44 | 0.93 | 0.28 | 0.61 |
| | $EF_{CRUSTAL}$ | 141 | 49 | 0.26 | 1.1 |
| | NSSF (%) | 0 | 74 | 97 | 25 |

561             Table 3: Crustal Enrichment Factors in rains of Abidjan, Lamto and Korhogo.

$SO_4^{2-}/Ca^{2+}$ ratio values for the three sites are higher than the reference ratio and their EF values are well
above 1. This result confirms that $SO_4^{2-}$ in rain could be explained by marine, crustal and some potential
additional sources. Marine contributions (SSF) of $SO_4^{2-}$ have been estimated in the range of 10 to 16%
along the transect Abidjan-Lamto-Korhogo and non-marine (NSSF) $SO_4^{2-}$ contributions are estimated to be
85 %, 90 %, 74 % for Abidjan, Lamto and Korhogo respectively. We hypothesize that urban site rainfall at
Abidjan and Korhogo could be influenced by $SO_4^{2-}$ of anthropogenic origin. $SO_4^{2-}$ and $NO_3^-$ often result





569 from anthropogenic emissions in urban areas (Keresztesi et al., 2020). $SO_4^{2-}$ and $NO_3^-$ concentrations in the

570 rainfall at the two urban sites are higher than those of Lamto (Table A1). In addition, we note that Abidjan

571 presents higher $SO_4^{2-}$ concentrations (19.50 µeq. $L^{-1}$) compared to the other two sites. In using the ratio

572 $SO_4^{2-}/NO_3^-$, we are able to distinguish between mobile sources such as traffic sources and stationary sources

573 such as industry (Xu et al., 2015; Keresztesi et al., 2019). The $SO_4^{2-}/NO_3^-$ ratio at Abidjan is 1.87, which,

574 given the urban context, suggests the leading role of stationary sources e.g., industry or charcoal-burning

575 emissions from domestic combustion (Li et al., 2020; Naimabadi et al., 2018). However, the fuel for vehicles

576 used in West Africa and particularly in Côte d' Ivoire, contains high levels of sulfur (Marc et al., 2016).

577 Thus, high VWM $SO_4^{2-}$ concentrations could be linked to traffic of motorized vehicles. In addition (Bahino

578 et al., 2018) have shown that $SO_2$ has three possible sources in Abidjan, i.e. traffic, domestic fire and waste

579 burning with traffic contributing the most.

580 Korhogo has a $SO_4^{2-}/NO_3^-$ ratio equal to 0.58 less than 1 illustrating the relative importance of $NO_3^-$

581 emissions compared to $SO_4^{2-}$. This result corroborates other rain chemical characteristics established at

582 Korhogo, which is considered a moderately industrialized city. In Abidjan, the importance of anthropogenic

583 emissions compared to the two other sites is explained by the level of population density, urbanization and

584 industrialization. Nevertheless, the $SO_4^{2-}$ concentration in Abidjan precipitation is lower by a factor 2 or 3

585 than those recorded in megacities such as Hong Kong, Jiaozhou bay (China) and New Delhi (India) (Wai et

586 al., 2005; Tiwari et al., 2007; Xing et al., 2017) (Table 4).
















| Sites | period | n | pH | H⁺ | Ca²⁺ | Mg²⁺ | Na⁺ | K⁺ | NH₄⁺ | HCO₃⁻ | Cl⁻ | SO₄²⁻ | NO₃⁻ |
|---|---|---|---|---|---|---|---|---|---|---|---|---|---|
| Limeira, Brazil[a] | 09/2013 –03/2014 | 30 | 5.62 | 2.40 | 54.88 | 17.40 | 22.39 | 5.68 | 34.36 | 20.13 | 7.06 | 15.54 | 14.73 |
| Jiaozhou Bay, China[b] | 06/2015 – 05/2016 | 49 | 4.77 | 16.90 | 64.10 | 21.90 | 54.7 | 17.20 | 107.00 | _ | 66.00 | 93.70 | 62.90 |
| Juiz de Fora, Brazil[c] | 2014 | 53 | 6.60 | 0.40 | 31.90 | 13.80 | 29.10 | 16.00 | _ | 8.50 | 18.30 | 3.00 | 25.60 |
| Lijiang City, China[d] | 06/2012 – 11/2012 | 176 | 6.07 | 0.85 | 50.10 | 10.90 | 0.98 | 2.01 | 20.80 | _ | 2.04 | 23.70 | 7.00 |
| Djougou, Benin[e] | 2006–2009 | 530 | 5.10 | 6.46 | 13.30 | 2.10 | 3.80 | 2.00 | 14.30 | _ | 3.40 | 6.20 | 8.20 |
| Florianópolis, Brazil[f] | 08/2006 – 11/2006 | 22 | 4.97 | 10.71 | 7.98 | 9.00 | 59.80 | 3.14 | _ | _ | 56.94 | 9.94 | 15.18 |
| Ibiúna, Brazil[g] | 2006 | 15 | 6.23 | 0.59 | 114.00 | 10.10 | 37.70 | 8.25 | 56.70 | _ | 21.20 | 60.90 | 21.80 |
| Delhi, Índia[h] | 2003–2005 | 355 | 6.39 | 1.02 | 80.88 | 23.11 | 24.35 | 14.18 | 31.81 | 38.42 | 29.52 | 40.81 | 25.17 |
| Porto Alegre, Brazil[i] | 2005–2007 | 177 | 5.30 | 4.98 | 22.40 | 9.28 | 18.40 | 6.48 | 35.30 | _ | 16.10 | 22.10 | 3.95 |
| Guaíba, Brazil[j] | 01/2002 – 12/2002 | 70 | 5.72 | 1.90 | 8.41 | 3.85 | 11.10 | 2.81 | 28.10 | _ | 6.98 | 13.20 | 2.47 |
| Ilha Grande, Brazil[k] | 03/2002 – 09/2002 | 20 | 5.22 | 6.00 | 9.20 | 40.40 | 142.20 | 7.10 | 9.90 | _ | 178.20 | 34.80 | 12.00 |
| São Paulo, Brazil[l] | 01/2003 – 12/2003 | 44 | 5.39 | 4.03 | 21.60 | 6.60 | 8.64 | 9.55 | 37.10 | _ | 9.29 | 23.80 | 20.10 |
| Ankara, Turkey[m] | 09/1994 -12/1996 | 162 | 6.33 | 1.60 | 71.4 | 9.30 | 15.60 | 9.8 | 86.40 | | 20.40 | 48.00 | 29.20 |
| Southern Taiwan[n] | 05/2005 – 12/2008 | 402 | *** | | 53.40 | 32.60 | 97.10 | 10.90 | 50.20 | 119.60 | 63.10 | 40.50 | 15.70 |
| Newark, USA[o] | 2006 – 2007 | 46 | 4.60 | 25.0 | 6.00 | 3.30 | 10.90 | 1.30 | 24.40 | _ | 10.70 | 38.10 | 14.40 |
| Hong Kong, China[p] | 10/1998 – 10/2000 | 156 | 4.20 | 63.20 | 16.20 | 7.00 | 36.90 | 4.20 | 22.00 | _ | 42.40 | 70 | 27.60 |
| Abidjan, Côte d'Ivoire[*] | 2019-2020 | 165 | 5.78 | 3.90 | 25.00 | 5.80 | 21.50 | 3.60 | 19.01 | 5.00 | 24.30 | 19.50 | 8.70 |
| Lamto, Côte d'Ivoire[*] | 2019-2020 | 146 | 5.31 | 6.57 | 9.91 | 2.57 | 5.41 | 2.00 | 17.90 | 1.50 | 5.57 | 4.76 | 7.22 |
| Korogho, Côte d'Ivoire[*] | 2019-2020 | 97 | 5.57 | 4.09 | 20.09 | 3.40 | 11.24 | 8.63 | 17.38 | 2.30 | 9.57 | 5.27 | 9.90 |


Table 4: Average VWM (µeq. L⁻¹) in rainwater in Abidjan, Lamto and Korhogo, Côte d' Ivoire (this study) and other
places in the world (adapted from (Martins et al., 2018)

The K⁺/Ca²⁺ ratios are below the reference crustal value and the EF values are above 1 for all three sites. In
addition, K⁺ and Ca²⁺ are highly correlated with r value of (r=0.79), (r=0.70), (r=0.73) (Figure 3)
respectively at Abidjan, Lamto and Korhogo. These results indicate a possible terrigenous origin of K⁺. The
NSSF K⁺ fraction was found to be 87%, 96% and 97 % at Abidjan, Lamto and Korhogo respectively (Table
4). However, a strong correlation coefficient between Cl⁻ and K⁺ with (r=0.75), (r=0.71) and r= (0.86) found
respectively at Abidjan, Lamto and Korhogo could suggest a potential K⁺ origin from biomass combustion,
which is a source of KCl (Lara et al., 2001). Since Submicron K⁺ is considered to be an atmospheric tracer
of biomass combustion (Andreae et al., 1998; de Mello, 2001), in the urban context of Abidjan and Korhogo
we can attribute household fire burning using charcoal as source of K⁺ whereas in Lamto biomass burning
of vegetation is likely to be the main source of K⁺.

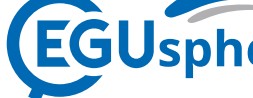

Calculations of $Mg^{2+}$marine, non-marine and crustal ratios (SSF and NSSF) and EFs indicate that the
terrigenous contribution of this ion is limited to 19% at Abidjan and 25% at Korhogo while it represents
65% at Lamto with a strong correlation between $Ca^{2+}$ and $Mg^{2+}$ (r=0.92). NSSF of $Ca^{2+}$ are 97 %, 98% and
98% respectively at Abidjan, Lamto and Korhogo. $Ca^{2+}$ displays high correlations with $SO_4^{2-}$ (r=0.86),
(r=0.76), (r=0.91), $Mg^{2+}$(r=0.81), (r=0.92), (r=0.91) and $K^+$ (r=0.79), (r=0.70), (r=0.73) respectively at
Abidjan, Lamto and Korhogo. We found that $Ca^{2+}$is the most important ion in the rainwater composition
measured at Abidjan and Korhogo with a concentration of 38.30 µeq. $L^{-1}$, 20.09 µeq. $L^{-1}$ respectively. At
Lamto it is the third most important ion with a concentration of 9.91 µeq. $L^{-1}$. The $Ca^{2+}$ predominance in
rain at the urban sites (Abidjan and Korhogo) could be explained by a contribution of multiple sources. In
Abidjan, the expansion of construction activities involving cement production represents a potential source
of $Ca^{2+}$ particles (Shakya et al., 2017; Samara et al., 2000; Khwaja et al., 1990). In addition, soil particle
resuspension from road dust can also contribute to the precipitation $Ca^{2+}$ content in precipitation (Tiwari, et
al., 2007 ; Fernandes et al., 2012 ; Kulshrestha et al., 2003 ; Riccio et al., 2017). To further investigate the
pattern of VWM $Ca^{2+}$concentration in rainwater, we analyzed Hysplit air mass back-trajectories at specific
dates (Abidjan: 10 April 2020, Lamto: 6 march 2020, Korhogo: 28 march 2019) representative of a
maximum $Ca^{2+}$ VWM concentrations of 555 µeq. $L^{-1}$, 231 µeq. $L^{-1}$ and 164 µeq. $L^{-1}$ for Abidjan, Lamto and
Korhogo respectively (Figure 6). At the transition between the dry and the wet season (march-April), we
observe that north east air masses coming from the Saharan desert at 2500 m of altitude are heavily loaded
with dust particles rich in terrigenous chemical components (such as $Ca^{2+}$) which affects all the three sites.
The scavenging of these particles by rainy events at the seasonal transition explains the magnitude of VWM
$Ca^{2+}$ concentrations recorded in rainwater at the three sites at the beginning of the wet season. with
concentrations ranged from 3.95 to 555 µeq. $L^{-1}$, 1.2 to 231 µeq. $L^{-1}$ and 1 to 164 µeq. $L^{-1}$ at Abidjan, Lamto
and Korhogo respectively.








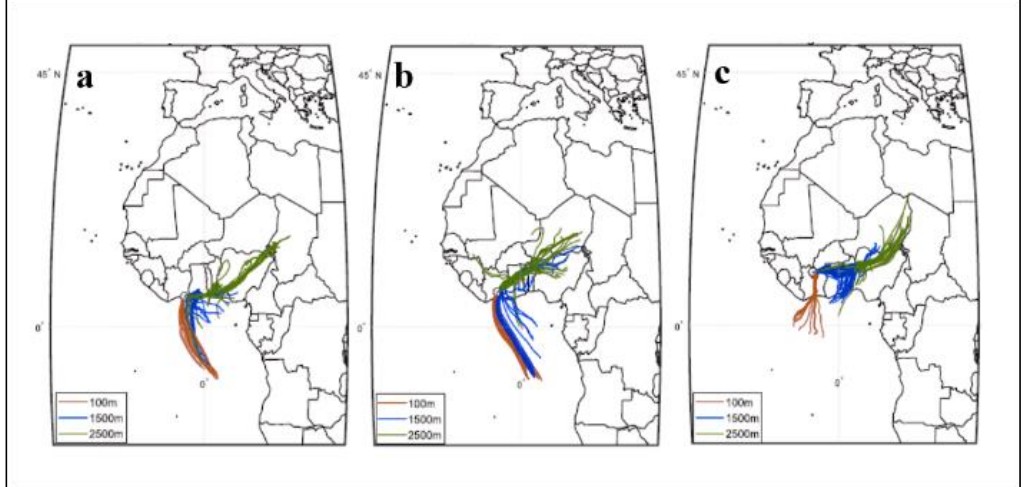

Figure 6: 96-hour overlay back-trajectories initiated in Abidjan (a: Abidjan: 10 April 2020), Lamto (b: 6 march 2020) and Korhogo (c: 28 march 2019)

The terrigenous signature identified at African sites emphasizes the direct influence of soil dust on rainfall (Laouali et al., 2021). The North African desert areas (Sahel and Sahara) are probably the most important mineral aerosol source (Kaufman, 2005; Marticorena et al., 2010) and when the monsoon sets in, North East Harmattan air masses heavily loaded in soil dust terrigenous components are transported over the continent. Due to the partial dissolution of soil dust, rain is loaded with dissolved calcium and carbonates (calcite), dolomite, gypsum and also illite, smectite or palygorskite which explains the enrichment of $Ca^{2+}$, $SO_4^{2-}$, $Mg^{2+}$ and $K^+$ (Avila et al., 1997).

This result is similar to those obtained in other African ecosystems (Galy-Lacaux and Modi , 1998; Sigha-Nkamdjou et al., 2003; Galy-Lacaux et al., 2009; Laouali et al., 2012; Akpo et al., 2015) and other regions of the world including Asia (Tiwari et al., 2016). At Korhogo, the traffic and industrial sources of terrigenous $Ca^{2+}$ are certainly lower compared to Abidjan. We presume that the $Ca^{2+}$ terrigenous contribution is primarily related to Saharan dust transport that is predominant at Korhogo located in the northern climatic zone influenced by warm and dry air masses (Harmattan) during the boreal winter (Marticorena et al., 2010) (Figure 5-h). Lamto rains present the lowest $Ca^{2+}$ VWM concentration (9.91 µeq. $L^{-1}$) over the studied transect and the lowest terrigenous contribution. This result is comparable with that of Yoboué et al., (2005) (9.20 µeq. $l^{-1}$) and may be explained by the position of Lamto, which is located in the climatic center zone and appeared to be less influenced by the harmattan air masses than Korhogo as shown by air mass back trajectories analysis (Figure 5-d,5-g).





$Ca^{2+}$ has a significant impact on the acidity neutralizing potential of precipitation and consequently it would
be useful to compare the concentration established in Côte d'Ivoire with concentrations in the rest of the
world (Table 4). Abidjan, Lamto and Korhogo record a $Ca^{2+}$ VWM concentration lower than cities such as
New Delhi (India) (80.88 µeq. $L^{-1}$), Limeira (Brazil) (54.88 µeq. $L^{-1}$) and Ankara (Turkey) (71.40 µeq. $L^{-1}$
$^{-1}$), that present a VWM concentrations ranging from 55 to 80 µeq. $L^{-1}$ of $Ca^{2+}$. However, $Ca^{2+}$ concentrations
in Côte d'Ivoire are higher than those of Florianopolis, Brazil (7.98 µeq. $L^{-1}$) and Newark (USA) (6.00 µeq.
$L^{-1}$). Thus, highly urbanized and industrialized cities with a dense demography, like the megacity of Abidjan,
will tend to have higher $Ca^{2+}$ and $SO_4^{2-}$ concentrations as a result of industrial activities, vehicle emissions
(including road resuspension) and other emissions related to urban activities (Hoinaski et al., 2013). The
medium-sized city of Korhogo, less urbanized and industrialized than Abidjan and the rural, wet-savanna
site of Lamto are more influenced by continental air mass transport from African deserts.
The contribution of terrigenous compounds is computed according to the equation: $NSS[Ca^{2+}] + NSS[Mg^{2+}]$
$NSS[K^+] + NSS[SO_4^{2-}] + Tcarb$. It is estimated to be 74.21 µeq. $L^{-1}$, 21.61 µeq. $L^{-1}$, and 37.25 µeq. $L^{-1}$ and
contribute to 39 %, 25%, 33% of the total ionic content respectively for Abidjan, Lamto and Korhogo. We
must specify that the so-called terrigenous contribution in this study represents a mixture of terrigenous and
anthropogenic sources in urban areas, whereas at the rural site of Lamto, it represents a mixture of
terrigenous and biomass burning sources (Figure 4). Abidjan and Korhogo present higher deposition fluxes
of terrigenous ionic compounds such as calcium with annual averages of $11.32 \pm 6.27$ kg $ha^{-1}$ $yr^{-1}$, and $4.50\pm$
2.07 kg $ha^{-1}$ $yr^{-1}$ respectively. $Ca^{2+}$ deposition fluxes in Lamto are lower with an annual average value of
$2.59 \pm 0.31$ kg $ha^{-1}$ $yr^{-1}$. This may reflect the difference between the urban site and rural site in term of acid
buffering capacities of rainwater with the urban sites of Abidjan and Korhogo more heavily affected by this
phenomenon than the rural site of Lamto.

3.1.3. Nitrogenous contribution
Nitrogenous contribution, defined as the sum of ammonium and nitrate VWM concentrations $[NH_4^+]$ +
$[NO_3^-]$, is respectively estimated to be 33.70 µeq. $L^{-1}$, 25.12 µeq. $L^{-1}$, and 26.47 µeq. $L^{-1}$, representing 18
%, 30 % and 25% of the total precipitation composition at Abidjan, Lamto and Korhogo respectively.
Abidjan's rainfall composition exhibits the weakest nitrogenous contribution. However, Abidjan records
the highest VWM concentrations of nitrogenous species with $NH_4^+$ VWM concentration value of 22.60 µeq.
$L^{-1}$ or 67% and $NO_3^-$ VWM concentration value of 11.10 µeq. $L^{-1}$ or 32 %.
In the context of urbanization and demographic growth, the development of fossil-fuel combustion from
road traffic and domestic combustion may play an important role in $NH_3$ and $NO_2$ emissions (Ehrnsperger
and Klemm, 2021). Similarly, (Bahino et al., 2018) state that $NO_2$ gas emissions in Abidjan have two




distinct sources: (i) the limited traffic of garbage collection vehicles, and the circulation of mini buses called
"Gbaka", which connect the city center to the suburbs, with average concentration levels of $17.8 \pm 4.7$ ppb,
and (ii) industrial activities, with average concentration levels of $20.9 \pm 2.8$ ppb. In contrast, $NH_3$ gas
emissions are strongly linked to biomass burning (firewood and charcoal) as a source of energy by most
households, with average concentration levels of $84.9 \pm 17.9$ ppb, and emissions from waste burning, with
average concentration levels of 39 ppb in Abidjan.
Lamto exhibits the highest nitrogenous contribution, representing one third of its rainfall composition,
consisting of 71% $NH_4^+$ and 29 % $NO_3^-$. The high $NH_4^+$ concentration could be explained by the rural
features of Lamto, such as bacterial decomposition of urea in animal excreta and emissions from natural or
fertilized soils by agriculture activities, which are both sources of $NH_3$ (Schlesinger and Hartley, 1992; Galy-
Lacaux and Modi, 1998). As noted by Suzanne et al., (2016), agricultural activity contributes to substantial
$NH_3$ gas emissions, resulting from the use of large quantities of fertilizers and plant phytosanitary product
to increase rubber and cocoa harvests around Lamto. Savanna fires and household fuelwood burning are
also primary sources of $NH_3$ (Delmas et al., 1995). Strong correlations of both $NH_4^+$ and $NO_3^-$ with $SO_4^{2-}$
(r= 0.72 and r=0.77 respectively) (Figure 3) indicate that $NH_4^+$ is related to multiphase reactions in the
atmosphere. Most of the time ammonia  exists in multiple form of aerosols in the atmosphere as $(NH_4)_2SO_4$,
$NH_4HSO_4$, $NH_4NO_3$ and $NH_4F$    (Seinfeld, 1986; Zhang et al., 2007a). The relatively low $NO_3^-$ VWM
concentration measured at Lamto could be explained by the low biogenic NOx emissions in the Lamto
station according to (Serça et al., 1998).  Further, (Akpo et al., 2015) propose that $NO_3^-$ in  rainwater could
be the result of gas-phase transformations of $NO_x$ to $HNO_3$, followed by a reaction with $NH_3$ to form
$NH_4NO_3$.
Nitrogenous concentration values measured at Lamto in this study for the period 2019-2020  are in the same
range as those found in the study of Yoboué et al., (2005) from 1995-2002  , with 17.6 µeq. $L^{-1}$ of $NH_4^+$and
7.7 µeq. $L^{-1}$ of $NO_3^-$. The nitrogenous contribution at the Korhogo site represents the second most important
contribution to the rain chemical content. $NH_4^+$ is dominant (71%) compared to $NO_3^-$ (29%). This result is
likely related to combined rural and urban characteristics of Korhogo, which allow a mixture of sources.
$NH_4^+$ concentrations are likely related to both the emissions of $NH_3$ from household charcoal burning
(Dentener and Crutzen ,1994; Zhang et al., 2007) , as well as biomass burning and the use of N-fertilizer in
agriculture around Korhogo (Galy-Lacaux and Modi, 1998; Laouali et al., 2012; Delmas et al., 1995). The
ratio $SO_4^{2-}/NO_3^-$ has a value of 0.53, which may indicate a substantial mobile source contribution (Rao et
al., 2017).It is worth recalling that  two-wheeled vehicles are predominant in Korhogo and could be a
possible source of $NO_2$ gases (Roger et al., 2016).VWM concentrations of nitrogenous species ($NH_4^+$, $NO_3^-$
) measured in this study are smaller than the values of cities such as Sao Paulo, Brazil (37.10, 20.00 µeq. $l^-$
$^1$); New Delhi, India (31.81, 25.17 µeq. $L^{-1}$); and Jiaozhou Bay, China (107, 62.90 µeq. $L^{-1}$), and are slightly
higher than the value of the West African rural site of Djougou in Benin (14.30, 8.20 µeq. $L^{-1}$) (Table 4).




Thus, the level of $NH_4^+$ VWM concentration in Côte d'Ivoire and in Benin are largely lower than the levels
recorded in urban areas such as in Brazil or in China where urbanization, fossil fuel consumption and
industrialization of agriculture (including intensive use of fertilizers and animals' manures) surroundings
cities are more significant (Migliavacca et al., 2005).
In the two rural wet savanna sites of Lamto and Benin, VWM $NH_4^+$ concentrations are attributed mainly to
livestock breeding, biomass burning, and, to some extent, agricultural activities (Zhang et al., 2007).
Nitrogen is considered to be an important source of nutrients in ecosystems, however, levels above a certain
critical load, which depends on the specific ecosystem, can be considered to be contributing to pollution and
eutrophication of the environment (Bobbink et al., 2010; Josipovic et al., 2011). We have calculated the
total annual nitrogen wet deposition fluxes for the three sites, finding values of 7.01 kg N ha$^{-1}$ yr$^{-1}$, 4.61 kg
N ha$^{-1}$ yr$^{-1}$, and 4.18 kg N ha$^{-1}$ yr$^1$ respectively for Abidjan, Lamto and Korhogo. We may conclude that
nitrogen wet deposition fluxes in the megacity of Abidjan are relatively more important than in the rural
area of Lamto and the regional/local connector city of Korhogo. In addition, we emphasize that annual
nitrogen wet deposition fluxes at the three sites are dominated by $N-NH_4^+$ with wet deposition fluxes of 4.68
kg N ha$^{-1}$ yr$^{-1}$, 3.27 kg N ha$^{-1}$ yr$^{-1}$, and 2.73 kg N ha$^{-1}$ yr$^{-1}$ respectively. $N-NO_3^-$ deposition flux values are
lower by a factor of two, with values of 2.33 kg N ha$^{-1}$ yr$^{-1}$, 1.34 kg N ha$^{-1}$ yr$^{-1}$, and 1.45 kg N ha$^{-1}$ yr$^{-1}$
respectively at Abidjan, Lamto and Korhogo. The values of nitrogen wet deposition remain lower than the
critical load, estimated to be 10 kg N ha$^{-1}$ yr$^{-1}$ , which is defined as the highest load that will not cause
chemical changes leading to long-term harmful effects on the most sensitive ecological systems (Nilsson,
1988; Bobbink et al., 2010). We can conclude that the three sites are not exposed to potential risks of
eutrophication or acidification of their environment for the moment. However, further investigations need
to be undertaken to assess total nitrogen deposition, including both wet and dry processes as well as the
evolution of those fluxes in the future.

### 3.1.4. pH and Acid contribution
Mean pH measurements for the three studied sites over the period 2018-2020 are given in Table A1 while
Figure 7 presents the pH frequency distribution.





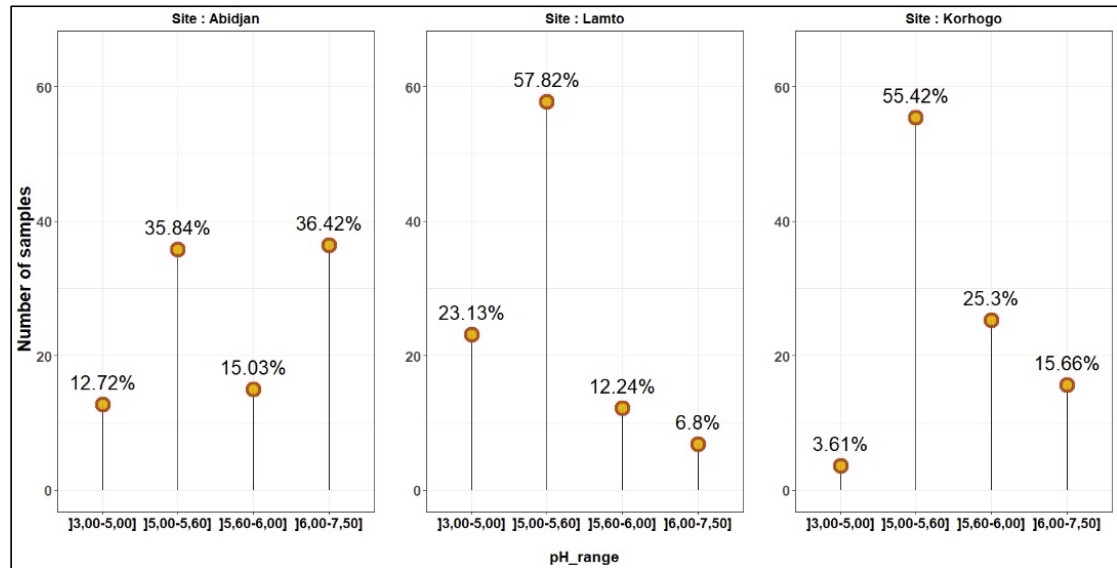

Figure 7: Frequency distribution of pH values of Abidjan, Lamto and Korhogo rainwater for the study period.

The mean pH is $5.76 \pm 0.59$, $5.31 \pm 0.32$ and $5.57 \pm 0.30$ respectively at Abidjan, Lamto and Korhogo. Mean VWM $H^+$ concentrations, calculated from annual mean precipitation pH are estimated to be $4.1 \pm 0.10$ µeq. $L^{-1}$, $6.57 \pm 0.04$ µeq. $L^{-1}$ and $4.09 \pm 1.7$ µeq. $L^{-1}$ at Abidjan, Lamto and Korhogo respectively (Table A1). The reference pH of rainwater is 5.6 representing the acidity of the pure water in equilibrium with the atmospheric $CO_2$ concentration (Charlson and Rodhe, 1982; Galloway et al., 1982). Acid rain is defined as rain with pH below the threshold of 5.60 (Drever et al., 1997). Results show that for the study period of 2018-2020, 51 % of the 173 rain events whose pH values were measured in Abidjan, present an acidic pH lower than 5.6, while 49% have an alkaline pH (>5.6). At Lamto, precipitation is mainly acidic with 81% of the 148 rainfall samples having a pH value measured below 5.6.

At Korhogo, precipitation was slightly acidic with 59% of the 83 rainfall samples having measured pH values below 5.6. This result is similar to the study of (Payus et al.2020) where rural areas recorded higher acidity of precipitation, with a total average pH of $5.54 \pm 0.39$ compared to urban areas with a total average pH of $5.77 \pm 0.26$. Abidjan precipitation presented a pH value close to cities such as Guiaba, Brazil (5.72), Lijiang city, China (6.07) and Ibuina, Brazil (6.23) and Korhogo has a pH value similar to cities such as Limeira (5.62) (Table 5). In comparison to others rural ecosystems, Lamto has a pH value close to Djougou (5.10) but lower than those of Sahelian sites such Katibougou (5.54), Dahra (6.10) and Agoufou (6.28)(Laouali et al., 2021).

Electrical conductivity (EC) of rainwater relies on total soluble components and lower EC values reflect better atmospheric environmental quality (Zhang et al., 2007). Mean EC values of precipitation measured





at Abidjan, Lamto and Korhogo ranged from 0 to 169 µS cm$^{-1}$, with means of 21.36 µS cm$^{-1}$, 6 µS cm$^{-1}$ and
5.9 µS cm$^{-1}$ respectively.  Based on this observation, EC rain characteristics emphasize lower environmental
quality of Abidjan, a polluted megacity compared to Korhogo and Lamto.
The contribution of mineral acidity, mainly related to the incorporation of $H_2SO_4$ and $HNO_3$ is 69%, 38%
and 52% respectively at Abidjan, Lamto and Korhogo while organic acids represent 31 %, 62% and 48 %
of acidity respectively at each site (Table 5). Abidjan and Korhogo are mainly influenced by mineral acidity
whereas Lamto show a high organic contribution (Table 5). The patterns observed at the urban sites are
comparable to measurements made in other African ecosystems, especially in South African dry savannas
influenced by anthropogenic sources (Mphepya et al., 2004; Conradie et al., 2016).
In figure 4, the organic contribution is computed according to the equation: $[SO_4^{2-}] + [NO_3^-] + [HCOO^-] +$
$[CH_3COO^-] + [C_2H_5COO^-] + [C_2O_4^{2-}]$, with values representing 7 %, 23% and 16 % of the source's
contributions to the chemical content of rainfall in Abidjan, Lamto and Korhogo respectively. The relatively
low organic acidity  in Abidjan is likely related to the fact that in urbans areas, anthropogenic activities  are
the main contributors of acid rain (Radojevic and Harrison , 1992; Park et al., 2015; Payus et al., 2020).
However, organic acids can be emitted in the urban environment, mainly from motorized vehicles. In the
study of  (Dominutti et al., 2019) in Abidjan , the majority of VOCs (Volatile organic carbon) which are
precursors of organic acids (Guenther, et al. 2013) are attributed to domestic fires, landfill fires and traffic
sources. At Lamto, the high acidity contribution confirms results previously established by (Yoboué et al.
2005) (56% organic and 44% mineral contributions). However, our study shows that the organic
contribution has increased compared to the period 1995-2002 studied by Yoboué et al., (2005).
The annual VWM concentration of $HCOO^-$, $CH_3COO^-$ and $C_2O_4^{2-}$ at Lamto are higher than at the urban
sites, especially for $HCOO^-$. The contribution of organic acidity in precipitation is mainly due to VOC
emissions from biomass burning and from the vegetation in rural sites (Guenther et al., 2006; Galy-Lacaux
et al., 2009; Vet et al., 2014). Despite acidity in Korhogo being dominated by the mineral acidity, there is
significant contribution from organic acidity. This configuration could be related to the singularity of the
urban site of Korhogo, which has a certain rural characteristic supporting a mixture of natural sources of
VOC, such as emissions from biomass burning and vegetation, and from anthropogenic sources such as
motorized vehicles. This result is similar to those found in the study of (Sun et al., 2016).
The neutralization Factor (NF) of mineral and organic acids by bases such as anions (oxides, carbonates or
bicarbonates, etc.) associated with base cations such as $Ca^{2+}$, $NH_4^+$, $Mg^{2+}$, $K^+$ can be evaluated by using  Eq
10 (Possanzini et al., 1988; Kulshrestha et al., 2003). Abidjan presents NF values for $Ca^{2+}$, $NH_4^+$, $K^+$, $Mg^{2+}$
of 0.87, 0.41, 0.10, and 0.21 respectively, revealing that $Ca^{2+}$ is the most important ion in neutralizing
acidity, followed by $NH_4^+$. At Korhogo and Lamto, both $Ca^{2+}$ (NF=0.64 and 0.32) and $NH_4^+$ (NF=0.57 and
0.58) ions are also the major ions that neutralized acids in rains. In Lamto, the importance of $NH_4^+$ as the



main neutralizing factor is emphasized but the strong correlations between $NH_4^+$ and organic acids ($HCOO^-$
, $CH_3COO^-$, $C_2O_4^{2-}$) respectively equal to r=0.92, r=0.90, and r=0.89.
Considering the equations of section 2.5, we calculated the Fractional Acidity (FA) to evaluate the
neutralizing strength of acidifying compounds Eq 9 (Balasubramanian et al., 2001). The average FAs for
the studied period (2019-2020) at Abidjan, Lamto and Korhogo are estimated to 0.11, 0.21 and 0.13. It
indicates that 89%, 79% and 87% of the rain acidity has been neutralized by alkaline substances at Abidjan,
Lamto and Korhogo respectively. According to (Sigha-Nkamdjou et al. 2003) who defined Potential acidity
(pA) as the sum of potential acidic compounds in the form of mineral and organic acids, we calculate pAs
equal to 44.11 µeq. $L^{-1}$ at Abidjan, 31.40 µeq. $L^{-1}$ at Lamto and 36.0 µeq. $L^{-1}$ at Korhogo. The measured
acidity (4.10 µeq. $L^{-1}$, 6.57 µeq. $L^{-1}$ and 4.09 µeq. $L^{-1}$) is lower by a factor of 3 to 10 compared to calculated
pA. In order to explain this gap, we assess the balance between neutralization and acidification processes in
rainwater chemistry by using Eq11. NP/AP values are respectively 1.36, 0.90, and 1.30 for Abidjan, Lamto
and Korhogo. Thus, we emphasize that neutralization processes are much more significant in the rainwater
chemistry of the urban sites of Abidjan and Korhogo compared to the rural site of Lamto (NP/AP<1).
Consequently, neutralization processes explain the differences between the potential $H^+$ and the measured
acidity in precipitation collected over Côte d'Ivoire's sites.
3.2. Monthly and seasonal concentration variation of major ions in rainwater

The monthly VWM ionic concentrations and WD are presented in Figure 8 (a-f) and Fig.A2. Results exhibit
seasonality at all three sites. During the dry season the ionic load are higher compared to the wet season.
Indeed, the first rains of the year (February-March in Abidjan, February in Lamto, March in Korhogo) at
each site show very high ionic load. The same result is obtained for the inter-seasons months from June to
September at Abidjan and for the last months of rain (November December) at all three sites. This observed
relationship between rainfall and VWM ions concentrations in rainwater is related to the atmosphere having
high levels of gas and particle scavenging at the beginning of the rainy season, but the gases and particles
are not highly diluted because of the small amounts of rain (Huang et al., 2009).






Figure 8: Monthly VWM concentrations of major ions (µeq. L$^{-1}$) at Abidjan (a), Lamto (b) and Korhogo (c) and monthly wet deposition (WD) fluxes (kgX.ha$^{-1}$. yr$^{-1}$) at Abidjan (e), Lamto (f) and Korhogo (g)






Monthly VWM concentrations variations of main ionic in Abidjan is dominated in the decreasing order by
$Ca^{2+}$, $Cl^-$, $Na^+$, $NH_4^+$, $SO_4^{2-}$ with values ranging respectively from 5.14 to 143.52 µeq. $L^{-1}$, from 4.01 to
166.47 µeq. $L^{-1}$, 3.72 to 185.79 µeq. $L^{-1}$, from 6.96 to 82.52 µeq. $L^{-1}$ and from 3.41 to 93.89 µeq. $L^{-1}$. Lamto
is dominated in the decreasing order by $NH_4^+$, $HCOO^-$, $Ca^{2+}$, $NO_3^-$, $Cl^-$ with values ranging respectively
from 6.14 to 146. µeq. $L^{-1}$, from 3.28 to 83.91 µeq. $L^{-1}$, from 2.42 to 108.14 µeq. $L^{-1}$, from 2.83 to 72.72
µeq. $L^{-1}$ and from 1.21 to 51.91 µeq. $L^{-1}$.
Korhogo presents in term of monthly VWM concentration the same dominants ions but with different order
of importance and so Korhogo is dominated in the decreasing order by $Ca^{2+}$, $NH_4^+$, $NO_3^-$, $HCOO^-$ and $Cl^-$
with values ranging respectively from 0.78 to 164.33 µeq. $L^{-1}$, from 3.61 to 44.74 µeq. $L^{-1}$, from 0.69 to
88.62 µeq. $L^{-1}$, from 0.25 to 52.53 µeq. $L^{-1}$ and from 1.09 to 65.98 µeq. $L^{-1}$. Conversely, Wet deposition (WD)
show an opposite trend compared to monthly VWM concentration distribution with maximum WD during
the wet seasons in all three sites (Figure 8). Monthly wet deposition fluxes at Abidjan are dominated by the
following ions in decreasing order $Cl^-$, $SO_4^{2-}$, $Ca^{2+}$, $NO_3^-$, $Na^+$ with values respectively ranging from 0.20
to 0.96 kg $ha^{-1}month^{-1}$, from 0.13 to 3.84 kg. $ha^{-1}month^{-1}$, from 0.14 to 6.33 kg $ha^{-1}month^{-1}$, from 0.07 to
3.12 kg $ha^{-1}month^{-1}$ and from 0.07 to 4.38 kg $ha^{-1}$ $month^{-1}$ (Figure 8).
Lamto displays dominant monthly wet deposition fluxes in decreasing order $HCOO^-$, $NO_3^-$, $NH_4^+$, $CH_3COO^-$
and $Ca^{2+}$ with values respectively range from 0.02 to 1.99 kg $ha^{-1}$ $month^{-1}$, from 0.05 to 1.20 kg $ha^{-1}$ $month^-$
$^1$, from 0.02 to 3.10 kg $ha^{-1}$ $month^{-1}$ from 0.03 to 1.57 kg $ha^{-1}$ $month^{-1}$ and from 0.02 to 4.39 kg $ha^{-1}month^{-1}$
respectively. Korhogo presents the same dominant ions in term of monthly wet deposition as Lamto with a
similar decreasing order: $HCOO^-$, $NO_3^-$, $CH_3COO^-$, $NH_4^+$, $Ca^{2+}$ and monthly wet deposition values are
ranged from 0.00 to 2.18 kg $ha^{-1}month^{-1}$, from 0.07 to 2.15 kg $ha^{-1}month^{-1}$, from 0.01 to 1.69 kg $ha^{-1}month^-$
$^1$, from 0.01 to 1.49 kg $ha^{-1}month^{-1}$ and from 0.04 to 1.51 kg $ha^{-1}month^{-1}$, respectively.
3.2.1 Seasonal and monthly concentration variation in Abidjan

On the Abidjan site, seasonal variation of the chemical signature of rainwater during the study period is well
marked. During the long dry season from December to February, rainwater chemistry is dominated by $Ca^{2+}$,
$SO_4^{2-}$, $NH_4^+$ ions with VWM concentration ranging from 31.60 to 81.67 µeq $L^{-1}$, from 19.6 to 49.54 µeq $L^-$
$^1$, and from 30.53 to 32.80 µeq $L^{-1}$ respectively. In the long-wet season from March to July the predominant
ions are $Na^+$, $Cl^-$ respectively ranging from 3.72 to 67.84 µeq. $L^{-1}$ and from 4.01 to 83.79 µeq. $L^{-1}$ (Figure
8-a). High concentrations during the dry season are due to dust particles coming from north east warm and
dry desert air masses (light blue and yellow cells) (Figure 5-b) and to gases emitted by biomass burning.  In
the wet season, Abidjan is under the influence of warm and humid monsoon air masses rich in sea salt which
explain the strong VWM concentrations of marine ions (Figure 5-c). The short dry season at Abidjan (from
August to mid-September) records VWM $Na^+$, $Cl^-$ concentrations ranging from 12.56 to 187.69 µeq. $L^{-1}$





and from 16.84 to 166.47 µeq. L$^{-1}$ respectively, showing the influence of monsoon air masses. During this
short dry season, coastal upwelling leads to a cooling of surface sea water which prevents the formation of
rain clouds due to a decrease in evaporation rate. The continental zone most impacted by this phenomenon
is the southeast of Ghana and extends towards the middle-east of Côte d'Ivoire where Abidjan is situated
(Ali et al., 2011). Finally, the short-wet season is marked by both marine ions Na$^+$ and Cl$^-$ with VWM
concentrations ranging from 9.86 to 38.57 µeq. L$^{-1}$, and from 12.11 to 77.76 µeq. L$^{-1}$ respectively. Alkaline
ions (Ca$^{2+}$, NH$_4^+$) VWM concentrations ranged from 10.60 to 100.21 µeq. L$^{-1}$ and from 13.52 to 53.30 µeq.
L$^{-1}$ respectively, highlighting a period of source mixing that indicates the end of the short-wet season and
the beginning of the long dry season.
Monthly VWM concentrations of major ions in Abidjan rains are on average 2 times higher in the long dry
season (December to February) than in the long- wet season (March to July). In the short-dry season,
monthly VWM concentrations are 3 times higher than in the short-wet season. Results indicate for the study
period (2018-2020) that the rain from the two dry seasons' rains represent only 12% of the total annual
rainfall and contributes to approximately 46% of the total measured chemical composition of precipitation.
3.2.2 Seasonal and monthly concentration variation in Lamto

At Lamto, rainwater chemistry is dominated by high monthly VWM NH$_4^+$ concentrations ranging from 6.14
to 146.58 µeq. L$^{-1}$ throughout the study period. During the long dry season from December to February,
chemical composition of rainwater is dominated by NH$_4^+$, Ca$^{2+}$, HCOO$^-$ and NO$_3^-$ with VWM concentration
value ranging from 28.20 to 146.58 µeq. L$^{-1}$, from 12.58 to 108.14 µeq. L$^{-1}$, from 8.14 to 83.91 µeq. L$^{-1}$ and
from 18.02 to 72.20 µeq. L$^{-1}$ respectively. The long-wet season is characterized by the same dominant ions
However, the level concentration of marine ions is higher than in the dry season. The influence of
nitrogenous and organic species in the dry season in Côte d' Ivoire  is related to periods of intense biomass
burning. Adon et al. (2010) reported maximum nitrogenous gases concentrations in West and central Africa
in the dry season months . Ossohou et al. (2019) confirmed this result and show that nitrogenous compounds
at Lamto are washed out by rainfall during the seven following months (April–October), with higher wet
deposition at the end of the dry season and at the beginning of the wet season, due to increased rainfall
during that period, when there is a strong influence of biomass burning (Figure 8-e). Monthly mean CrIS
NH$_3$ and OMI NO$_2$ vertical column densities (VCDs) and MODIS burned area fraction from January 2018
to December 2020 confirm that the highest NH$_3$ and NO$_2$ concentrations are related to active emission
sources in the dry season at all sites, but especially in Lamto (Figure 9). At Lamto, the highest monthly NH$_3$
and NO$_2$ VCDs are associated with the highest monthly burned area fractions, with monthly means ranging
respectively from 2.42 x 10$^{16}$ to 4.53 x 10$^{17}$ molecules NH$_3$ cm$^{-1}$, 1.32 to 12.91 x 10$^{15}$ molecules NO$_2$ cm$^{-2}$
and from 0.75 to 35.91% area burned, highlighting the importance of biomass burning source in Lamto
compared to the two studied urban sites. During the dry season, Lamto is also under the influence of

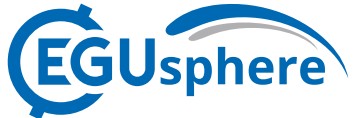

harmattan air masses loaded with dust, which may explain the level of VWM $Ca^{2+}$ concentration (Figure 8-
a) (Figure A2).















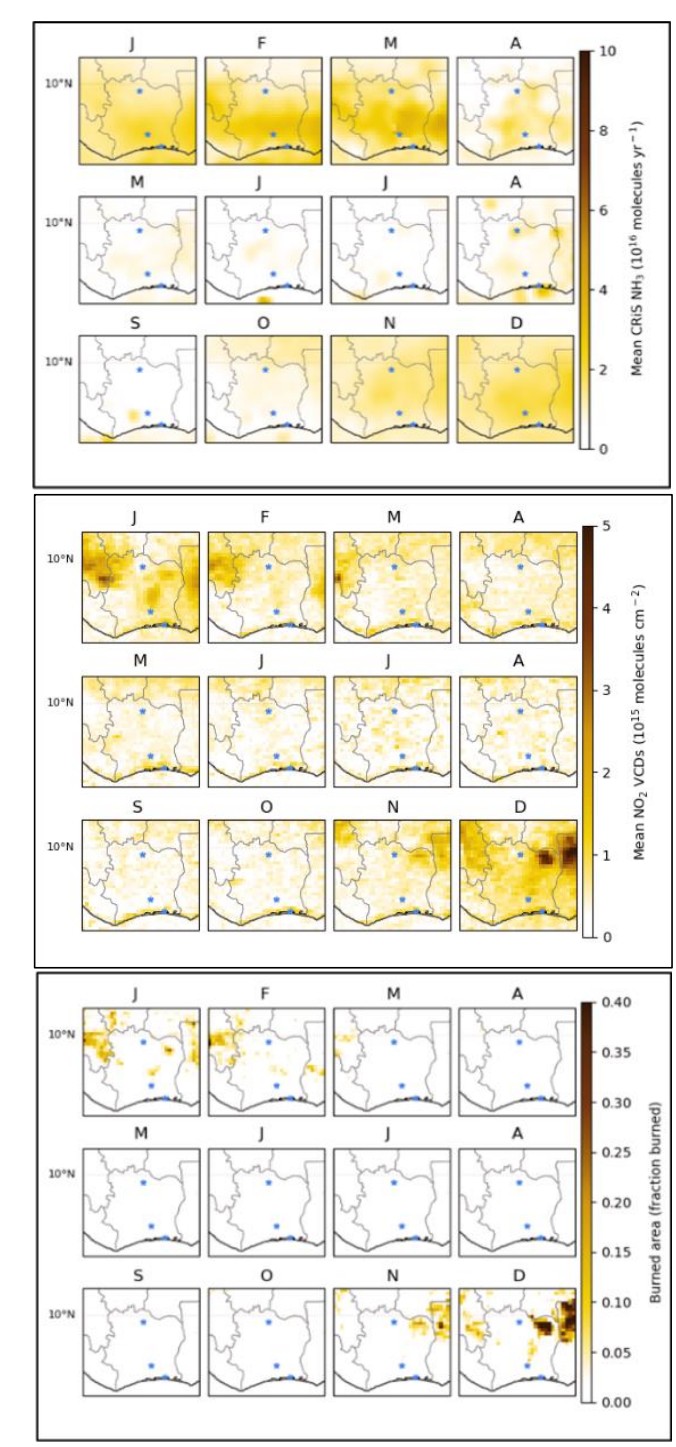

Figure 9: Monthly mean vertical column densities of $NH_3$ from CRiS (let panel), tropospheric vertical column densities of $NO_2$ from OMI (middle panel), and burned area from MODIS (right panel) averaged over 2018-2020. Note that the means for $NH_3$ do not include April-July 2019.





The marine ion and organic acid signature of the long-wet season is associated with the influence of oceanic
air masses on Lamto during the long-wet season (Figure 5-f). The short dry season restricted to August at
Lamto is significantly influenced by the marine source because of the same phenomenon described for the
short dry season in Abidjan. During the short-wet season from September to November, rainwater chemistry
is characterized by relatively high levels of concentrations of $Ca^{2+}$ and $NH_4^+$ and with a signature of marine
ions and organic acids at the end of this season. In Lamto for the study period (2018-2020), monthly VWM
concentrations of major ions show that concentrations are on average 4 times higher in the long dry season
(December to February) than in the long-wet season (March to July) while concentrations in the short-dry
season (August) are 3 times higher than in short-wet season (October to November). Rain from the dry
seasons represent only 3 % of the total annual rainfall but approximatively 74% of the total measured
chemical composition of precipitation in Lamto
3.2.3 Seasonal and monthly concentration variation in Korhogo

At Korhogo, the average seasonal evolution in Korhogo is marked by a predominance of $Ca^{2+}$, $NO_3^-$, $SO_4^{2-}$
and $NH_4^+$ ions during the long dry season from November to March and a long-wet season from April to
October, which presents rainwater chemical composition mainly dominated by the previously cited ions
dominant in the long-dry season but with much lower VWM concentrations due to the dilution effect. In
addition, in this long-wet season there is a signature of $HCOO^-$, $Na^+$, and $Cl^-$ ions in rains.
Terrigenous compound abundance is related to the ITCZ position, which places Korhogo under the
influence of dry and warm harmattan air masses. The high concentrations of nitrogenous compounds reflect
on the one hand, the strong agricultural activity in the surroundings of the city as well as the use of fertilizer
and biomass fires by farmers for field clearing before the wet season and on the other hand the relative
importance of traffic source in $NO_2$ emissions from motorized vehicles. Satellite data analysis show that
nitrogenous species reach their highest VCDs values in March-April at the end of the dry season at Korhogo;
values ranged respectively from $2.74 \times 10^{16}$ to $2.14 \times 10^{17}$ molecules $NH_3$ $cm^{-1}$, from $2.00$ to $8.54 \times 10^{15}$
molecules $NO_2.cm^{-2}$ and from 0.11 to 2.68% of burned area. The relative importance of $SO_2$ in dry seasons
is likely to be related to biomass burning (burning of forest, grassland, and agricultural wastes), which could
be also a significant source of $SO_2$ to the atmosphere which is more active in dry season than in wet
season(Bates et al., 1992; Arndt et al., 1997).
The increase in soil moisture at the beginning of the wet season is correlated to the increase of NO and $NH_3$
concentrations in the dry savanna (Adon et al., 2010) and could explain a significant part of $NH_4^+$ and $NO_3^-$
concentration at Korhogo. The growing vegetation in the wet season produces biogenic VOC emissions
which are an important source of organic acids, and can explain the preponderance of $HCOO^-$ in the wet
season at Korhogo, similarly to the result observed in Lamto. (Niu et al., 2018) found similar results in





China with highest acid concentrations in rains during the growing season (wet season) than in the non-
growing seasons (dry season). Finally for the study period (2018-2020) in Korhogo, monthly VWM
concentrations of major ions show that concentrations are on average 3 times higher in the long dry season
(November to March) than in the long-wet season (April to October). The dry season's rains represent only
3 % of the total annual rainfall, but approximatively 82% of the total measured chemical composition of
precipitation in Korhogo.

4.Conclusion
This paper documents rain chemical composition and associated wet deposition of major ionic species along
a North-South transect in Côte d'Ivoire. This study presents original results over a three-year time period at
two urban sites (Abidjan and Korhogo) inter-compared with a rural wet savanna site (Lamto) in Côte
d'Ivoire. The mean precipitation chemical content and wet deposition fluxes are computed at the annual
(2019 and 2020) and at the monthly scale for the full studied period (2018-2020). The dominant ion at both
urban sites is $Ca^{2+}$, whereas $NH_4^+$ dominates the chemical content of the Lamto rural site. At Abidjan and
Korhogo, $Ca^{2+}$, $Na^+$, $Cl^-$, and $NH_4^+$ and $Ca^{2+}$, $NH_4^+$, $Na^+$, and $HCOO^-$, respectively dominate the total
chemical content and represent 62% and 63% of the total. The rainwater chemical signature at the rural site
of Lamto is dominated by $NH_4^+$, $HCOO^-$, $Ca^{2+}$, and $NO_3^-$ ions, representing 55 % of the total. The two urban
sites of Abidjan and Korhogo rains are characterized by a terrigenous contribution associated to a mixture
of terrigenous continental and anthropogenic sources respectively 39% and 33%, also a high marine
contribution respectively 34 % and 24% and a significant nitrogenous contribution respectively 18 % and
25 % mainly associated to fossil fuel from road traffic, domestic and biomass burning sources. At the rural
Lamto site, marine, terrigenous and nitrogenous contribution represent respectively 14%, 25% and 30%. In
addition to the high nitrogenous contribution related to biomass burning and agricultural sources, the site
shows a strong organic component (23%) comparable to the one of Korhogo (16%). This original result has
been associated to volatile organic compounds emissions from biomass burning and vegetation at the rural
site and from domestic and landfill fires and traffic at the urban sites. Mean measured pH are respectively
5.76, 5.31, and 5.54 for Abidjan, Lamto and Korhogo indicating the importance of neutralization processes
in urban rainwater chemistry. Considering the eutrophication of the environment through rainwater,
significant wet deposition fluxes of nitrogen have been measured and equal respectively 7.01 kgN.ha$^{-1}$. yr$^{-1}$
$^1$, 4.61 kgN.ha$^{-1}$. yr$^{-1}$, 4.18 kgN.ha$^{-1}$.yr$^1$ in Abidjan, Lamto and Korhogo. This unique study on rainfall
composition in urban sites represents a first step to characterize urban deposition fluxes and to understand
the composition of the atmosphere and the pollution levels in African cities.  Quantifying urban key
deposition species such nitrogen is important for closing the gap in regional budgets of ionic species, which
are necessary for policy makers to manage atmospheric inputs to and outputs from local ecosystems,
assuming that a portion of total emitted urban compounds are transported out of the city into the surrounding



region. However, it is important to note that wet deposition studies should be complemented by dry
deposition processes evaluation to assess the global deposition budgets to improve knowledge on the
biogeochemical cycle balance, soil quality, water quality, and to improve global deposition models. There
is a clear need for more long term, quality-controlled in situ measurements in African urban areas, Africa
being a key continent in the future considering climate and environmental issues where national and
international pollutants reduction directives should be taken.

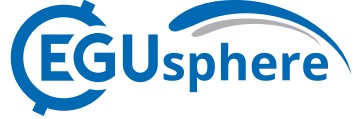


Appendices


| Species | ABIDJAN | | | | | | LAMTO | | | | | | KORHOGO | | | | | |
|---|---|---|---|---|---|---|---|---|---|---|---|---|---|---|---|---|---|---|
| | 2019 | | 2020 | | 2019-2020 | | 2019 | | 2020 | | 2019-2020 | | 2019 | | 2020 | | 2019-2020 | |
| | VWM | WD | VWM | WD | VWM | WD | VWM | WD | VWM | WD | VWM | WD | VWM | WD | VWM | WD | VWM | WD |
| H⁺ | 4.00 | 0.05 | 4.14 | 0.07 | 4.1(±0.09) | 0.06(±0.01) | 6.54 | 0.10 | 6.62 | 0.07 | 6.57(±0.06) | 0.09(±0.02) | 2.59 | 0.03 | 5.65 | 0.06 | 4.09(±2.16) | 0.05(±0.02) |
| pH | 5.57 | | 5.89 | | 5.76 | | 5.24 | | 5.38 | | 5.31 | | 5.70 | | 5.40 | | 5.57 | |
| Na⁺ | 19 | 5.93 | 30.95 | 11.34 | 26(±8.43) | 8.8(±3.82) | 4.46 | 1.55 | 6.82 | 1.73 | 5.41(±1.67) | 1.62(±0.13) | 16.65 | 4.45 | 2.40 | 0.60 | 11.24(±10.08) | 3(±2.83) |
| NH₄⁺ | 25.19 | 6.16 | 20.7 | 5.94 | 22.6(±3.18) | 6(±0.15) | 16.47 | 4.48 | 20.01 | 3.97 | 17.9(±2.50) | 4.20(±0.36) | 18.40 | 3.86 | 16.55 | 3.23 | 17.38(±1.31) | 3.50(±0.54) |
| N in NH₄⁺ | 19.64 | 4.8 | 16.14 | 4.63 | 17.63 | 4.68 | 12.84 | 3.49 | 15.6 | 3.09 | 13.96 | 3.27 | 14.35 | 3.01 | 12.90 | 2.51 | 13.55 | 2.73 |
| K⁺ | 3.78 | 2 | 5.09 | 3.17 | 4.5(±0.93) | 2.62(±0.83) | 1.80 | 1.06 | 2.8 | 0.99 | 2(±0.35) | 1.02(±0.05) | 12.43 | 5.65 | 2.06 | 0.87 | 8.63(±7.34) | 3.80(±3.52) |
| Ca²⁺ | 24.19 | 6.57 | 48.35 | 15.44 | 38.3(±17.08) | 11.32(±6.27) | 7.93 | 2.40 | 12.85 | 2.84 | 9.91(±3.48) | 2.59(±0.31) | 24.08 | 5.61 | 13.27 | 2.88 | 20.09(±7.64) | 4.50(±2.07) |
| Mg²⁺ | 5.49 | 0.9 | 8.74 | 1.67 | 7.4(±2.30) | 1.31(±0.55) | 2.19 | 0.40 | 3.12 | 0.42 | 2.57(±0.65) | 0.40(±0.01) | 4.07 | 0.57 | 2.35 | 0.31 | 3.4(±1.21) | 0.50(±0.20) |
| NO₃⁻ | 8.79 | 7.39 | 12.77 | 12.62 | 11.1(±2.81) | 10.16(±3.70) | 6.26 | 5.86 | 8.63 | 5.89 | 7.22(±1.67) | 5.84(±0.03) | 10.72 | 7.73 | 6.97 | 4.68 | 9.09(±2.65) | 6.30(±2.34) |
| N in NO₃⁻ | 2.02 | 1.69 | 2.93 | 2.90 | 2.55 | 2.33 | 1.43 | 1.34 | 1.98 | 1.35 | 1.66 | 1.34 | 2.46 | 1.77 | 1.60 | 1.07 | 2.09 | 1.45 |
| Cl⁻ | 24.40 | 11.74 | 37.44 | 21.18 | 32(±9.22) | 16.77(±6.68) | 4,.76 | 2.55 | 7.27 | 2.84 | 5.77(±1.77) | 2.67(±0.21) | 13.71 | 5.66 | 2.44 | 0.94 | 9.57(±7.94) | 3.80(±3.48) |
| SO₄²⁻ | 19.49 | 12.69 | 19.38 | 14.83 | 19.5(±0.08) | 13.76(±1.52) | 4.24 | 3.07 | 5.54 | 2.93 | 4.76(±0.92) | 2.99(±0.10) | 6,01 | 3.35 | 3.84 | 2.00 | 5.27(±1.53) | 2.80(±1.04) |
| S in SO₄²⁻ | 6.43 | 4.18 | 6.39 | 4.89 | 6.43 | 4.5 | 1.39 | 1.01 | 1.82 | 0.96 | 1.57 | 0.98 | 1.98 | 1.10 | 1.26 | 0.66 | 1.73 | 0.92 |
| *tCarb | 7.17 | 2.99 | 17.04 | 16.57 | 12(±4.01) | 11(±5.51) | 2.29 | 1.73 | 3.50 | 1.93 | 2.78(±0.80) | 2.21(±0.64) | 5.17 | 0.06 | 2.52 | 0.67 | 4.54(±1.87) | 3.10(±0.19) |
| HCOO⁻ | 5.57 | 3.48 | 7.76 | 5.70 | 6.80(±1.55) | 4.65(±1.57) | 9.74 | 6.76 | 14.12 | 7.16 | 11.51(±3.10) | 6.91(±0.28) | 8.71 | 4.66 | 11.18 | 5.57 | 9.97(±1.75) | 5.20(±0.53) |
| CH₃COO⁻ | 3.21 | 2.57 | 6.72 | 6.33 | 5.30(±2.48) | 4.57(±2.66) | 5.40 | 4.81 | 8.47 | 5.51 | 6.64(±2.17) | 5.11(±0.49) | 5.16 | 3.54 | 5.51 | 3.52 | 5.61(±0.24) | 3.70(±0.10) |
| C₂H₅COO⁻ | 0.00 | 0.00 | 0.00 | 0.00 | 0.00 | 0.00 | 0.08 | 0.00 | 0.14 | 0.00 | 0.10(±0.04) | 0.00(±0.00) | 0.00 | 0.00 | 0.00 | 0.00 | 0.00 | 0.00 |
| C₂O₄²⁻ | 4.08 | 1.69 | 1.28 | 1.83 | 1.4(±0.24) | 1.89(±0.60) | 0.94 | 0.64 | 1.41 | 0.70 | 1.13(±0.33) | 1.33(±0.04) | 3.09 | 1.62 | 0.48 | 0.39 | 2.09(±1.85) | 1.10(±0.19) |


Table A1: Annual Volume Weight Mean (VWM) (µeq L⁻¹) and wet deposition fluxes (WD) (KgX.ha⁻¹. yr⁻¹) in
rainwater in Abidjan, Lamto and Korhogo, (Côte d' Ivoire).










Figure A2: Monthly VWM (µeq. L[-1]) variation of major ions in rainwater at Abidjan (a, b, c, d); Lamto (e, f,
g, h) Korhogo (I,j,k,l)







| Acronyms | Signification |
|---|---|
| AII | Annual Inter- variability Index |
| AP | Acidification Potential |
| EC | Electrical Conductivity |
| EF | Enrichment Factor |
| FA | Fractional Acidity |
| GAW | Global Atmospheric Watch |
| NF | Neutralization Factor |
| NP | Neutralization Potential |
| NSSF | Non-Sea salt Fraction |
| OMI | Ozone monitoring Instrument |
| pA | Potential Acidity |
| PC | Percentage Coverage |
| Pc | Precipitation Collected |
| PCL | Percentage Coverage Length |
| Pt | Annual Precipitation |
| SSF | Sea-Salt Fraction |
| VCD | Volume Column Density |
| WD | Wet Deposition |
| VWM | Volume Weighted Mean |
| WMO | World Meteorological Organization |


1086        TABLE A3: List of acronyms

Data availability
Raw data were collected in the framework of the PASMU project and are available on request from the
coordinator Pr V. Yoboué (yobouev@hotmail.com). Data of the LAMTO site are available from the
INDAAF project at the address http://indaaf.obs-mip.fr  The pre-processed HYSPLIT trajectory data can
be obtained from the corresponding author, and the trajectories can be freely calculated at the web
page https://www.ready.noaa.gov/HYSPLIT_traj.php.  MODIS burned-area data are available from
https://doi.org/10.5067/MODIS/MCD64A1.006 (Giglio et al., 2015). OMI L3 NO SP version 4 is
available at 10.5067/MEASURES/MINDS/DATA301 (Lamsal et al., 2014). The CrIS CFPR version 1.6.3
NH3 VCD data is available upon request from Environment and Climate Change Canada
(mark.shephard@ec.gc.ca) https://hpfx.collab.science.gc.ca/~mas001/satellite_ext/cris/snpp/nh3/v1_6_3/
(Shephard et al., 2020).

Competing interests
The authors declare that they have no conflict of interest
Acknowledgements
This work carried out in part within the framework of the PASMU project, financed by the Ministry of
Higher Education and Research of Côte d'Ivoire, in the framework of Debt Reduction-Development
Contracts (C2D) managed by the Research Institute for Development (IRD) under funding program
5768A1-PRO/PRESED-CI C2D PASMU project whose main agreement is registered under number



305984/00. This work also received a contribution from the INDAAF program, supported by the
INSU/CNRS, the IRD (Institut de Recherche pour le Développement), from the Observatoire des Sciences
de l'Univers EFLUVE, from the Observatoire Midi-Pyrénées and the European Union's Horizon 2020
research and innovation program under Marie Skłodowska-Curie grant agreement No. 871944. The
authors would like to thank the PI and staff of the Lamto station in Côte d'Ivoire, SODEXAM (Airport,
Aeronautical and Meteorological Development and Exploitation Company), EVIDENCE project (Extreme
rainfall events, vulnerability to flooding and to water contamination) and finally the managers of the urban
site located at the IRD within the FELIX HOUPHOUET BOIGNY University in (Abidjan) and the second
urban site located at the PELOFORO GON COULIBALY University (Korhogo). We thank also Enrico
Dammers and Mark Shephard for their assistance with CrIS $NH_3$ data.

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
