# Peer review of "The chemical characteristics of rainwater and wet atmospheric deposition fluxes at two urban sites and one rural site in Côte d' Ivoire."

_EGUsphere, 2022_

## Author Comment (AC1)

**REVIEWER COMMENTS TO AUTHOR**

The authors thank the reviewers for their useful comments. We recognize that some manuscript sections were too long and repetitive. We improved the manuscript by reducing and synthesizing all the sections. In addition, we added a co-author (C. Mouchel-Vallon) to this paper who helped significantly in this review process, especially in the acid contribution section, and in the global rewriting of the paper.

Answers and explanations are given here, accompanied by detailed descriptions of the modifications brought to the manuscript. As suggested by Reviewer #1, we moved some content to newly created appendix and supplementary material.

Additionally, we revised the abstract and the conclusion to better reflect the content of the paper.

**Reviewer#1**

**General comments**

*General comment 1*: The paper is too long and very descriptive, sometimes repeating the details several times. This made the manuscript hard to read. The results section should be shortened, and the conclusions should be more explicit. I would suggest reducing the total manuscript's content by at least 40-50%. Please, move some information and details to supplementary material.

*Response to general comment 1*:

We have reduced drastically the manuscript. We rewrote some parts of text; we sent some parts of text and figures and tables in supplementary materials and we deleted a section.

We deleted the section Monthly and Seasonal Variations (195 lines) now integrated in the results sections presenting all the contributions.

Globally, the paper previously represented 826 lines against 675 lines in this reviewed version. The Material and Method section has been reduced from 301 lines to 192 lines. The results section has been reduced 408 lines to 366.

*General comment 2*: The authors could explore the application of statistical dimension reduction techniques, such as partial least squares or principal component analysis, which could help to improve the results by reducing individual analysis and the conclusions of the study.

**Response to general comment 2:**

We have performed the requested PCA analysis. Results are presented at the end of this document. As major findings don't constitute an added value compared to our calculations of correlation coefficient, we decided not to include this ACP analysis in the text.

*General comment 3:* The quality of the figures is not good enough. The authors should improve them by increasing the quality resolution and the size of the text used. Additionally, the captions need to be more detailed.

**Response to general comment 3:** We have improved quality resolution and the size of the text used in figures and try to detail more specifically captions.

*General comment 4:* References need to be incorporated in the discussion of the results.

**Response to comment 4:**

We have added references in the discussion part and this can be seen in revised manuscript:

Line 138

Line 518

**General comment 5**: A discussion about the limitations of the study is missing. Even if there is a lack of data in the continent, and some points are mentioned at the end of the conclusions, some discussion about the limitations and the improvements for future studies should be addressed.
**Response to general comment 5** We have addressed this issue in the conclusion.

**General comment 6:** The abstract and conclusions need to be improved. They are repetitive and seem more like a summary than proper abstract/conclusions
**Response to general comment 6:** We have improved abstract and conclusions.

**Minor comments reviewer#1**

Line 27: VWM, please add first the definition of the abbreviation
Now line 30: VWM has been defined, we write Volume Weighted Mean (VWM)

Figure 1: The resolution and text size are not good enough. Please improve it.
We modified Figure 1 to improve the quality

Lines 68-72: Too long. Please split the sentence.
We split the sentence as followed: *"In this context, the most recent study is the global assessment of precipitation chemistry and deposition carried out under the auspices of the World Meteorological Organization (WMO) - Global Atmospheric Watch (GAW) program. It aims to characterize precipitation chemical composition and to quantify deposition fluxes (wet, dry, total) of sulfur, nitrogen, acidity, sea salt, organic acids and phosphorus at global and continental scales."* Now line 73-77

Line 73: delete – before 2005 Done

Lines 94-99: Too long. Please split the sentence.
We modified the sentence to split and shorten as followed: *"In the context of the rapid urbanization and demographic explosion in Africa, it is important to improve the understanding of urban atmospheric composition and the potential impacts of air pollution in developing countries' megacities (United Nations, Department of Economic and Social Affairs, Population Division 2017; Kaba et al. 2020)"*. Now line 97-100

Lines 107-111: Too long. Please split the sentence.
The sentence has been shortened considering that all information for the programs are included in Gnamien et al, 2021 and in the INDAAF web site. *"This work was carried out within the framework of the Air Pollution and Health in Urban Areas program (PASMU) implemented in 2018 (Gnamien et al., 2021); and the INDAAF program"*. Now line 108 -110

Lines 112-114: Redundant, already said in lines 105-107
The sentence has been removed and some elements complete now the sentence in lines 105-107.

Line 138: delete ( before Fall Done

Lines 137-143: Hard to read; please reformulate

Following the comment of reviewer 2, we decided to remove the paragraph that defined connectors. In the aim to globally shorten the paper, we decided that the information given in lines 137-143 was not absolutely necessary.

Line 174: and elsewhere, Intertropical convergence zone (ITCZ)
OK, Line 174 (now line 258) we wrote Intertropical convergence zone (ITCZ) and then we just use the abbreviation ITCZ

Line 178: avoid repetition, the Northern, the Central and Coastal climatic zones Ok done, now line 262

Figure 2 and Lines 221 -271: This section describes observational data and part of the results. Please move to the results section or supplementary information. Also, quite repetitive about the climatic zones. That was already said in the previous paragraph. Please refer to each panel when describing Figure 2.
We decided to modify section 2: Material and Methods. Especially we modified section 2.2 now intitled Meteorological parameters (now 164 ). This section now only presents database and references for the acquisition of meteorological parameters.

We move the description of general climatology to the result section (section 3) and we add section 3.1 intitled "Climatology of sites" to present results of the Figure 2 (see now lines 256 - 330). This paragraph has been entirely modified.

Lines 239-240. Observations in Abidjan show a weak fluctuation of air temperature and relative humidity during the studied period (2018-2020). But the authors are only showing the data from 2019 in Figure 2. Please correct or explain.
We modified the sentence to explain that meteorological observations presented in Figure a,b,c for the three sites are a mean for the period 2018-2019 (see 305-309 ) *"Observations in Abidjan show a weak fluctuation of mean air temperature and relative humidity over the period 2018-2020".*

Section 2.3: is quite long. Please reduce.
We tried to shorten the section, but the definition and calculation of WMO/GAW criteria are important to explain the representativity of the study's sampling and finally of our database.

Section 2: Please define organic ions.
We write now line 218-219 *"Major inorganic (Na$^+$, K$^+$, Mg$^{2+}$, Ca$^{2+}$, Cl$^-$, NO$_3^-$ SO$_4^{2-}$, NH$_4^+$) and organic, derived from carboxylic acids, (HCOO$^-$, CH$_3$COO$^-$, C$_2$H$_5$ COO$^-$, C$_2$O$_4^{2-}$) ions were determined by Ionic Chromatography…"*

Line 354: NO2 Done

Line 366. Please define VCDs Done *"Vertical Column Densities (VCDs)"*. Now line 237

Line 387: Crustal element of continental origin?
We change in supplementary material in section 4 by "*Ca$^{2+}$ is selected as a reference element from crustal origin*"

Line 391, please define NSS before the equation.

We define SSF and NSSF now in supplementary material in (See S4) (previously line 390): *"Sea Salt Fraction (SSF) and Non-Sea Salt Fraction (NSSF)"*.

Lines 393 and 401: The authors use potential acidity and acidic potential… are those the same? This part is not clear; please organize it better and leave some spaces between equations and text.
We re-organized this part now and we decided to place this part in the supplementary material (See S4).

Figures 3 and 4 present a very low-resolution quality. Please improve them: Figure 3 is now in supplementary material noted Figure S1, Done

Line 475 and elsewhere. Terrigenous origin> Do the authors mean mineral dust /crustal origins? yes, Terrigenous origin we refer to mineral dust/crustal origin for this line but we have mentioned in legend that Terrigenous in urban context refers to a mixture of anthropogenic sources and crustal source.

Figure 4. The authors are comparing sources with chemical contributions in this figure. This is confusing. I do not think it is appropriate to use this classification, since one can have organic species from terrigenous or marine origin. The same is true for acidity. Also, it is not clear which ions are represented in each category.
This classification has been used in several studies (Laouali et al., 2012, 2021) ,(Bakayoko et al., 2021; Akpo et al., 2015). Moreover, we have specified in the paper that we are not comparing sources and contributions. We are trying evaluate the contribution of each group of ionic species that can come from the same source and to do this we have specified marine and non-marine fractions of the ionic species that enter in the calculation of contributions. Thus, terrigenous contribution is made up exclusively of non-marine fraction of ionic species that are recognized as having crustal origins. But as we refer to the non-marine fraction it includes the crustal and anthropogenic fraction that is why we have specified in the legend that the terrigenous contribution is a mixture of crustal and anthropogenic fraction. We will put the contributions and the different species for each contribution in the legend to make it clearer.

Line 495 and figure 5: Why are the authors using an extended period for back trajectories when the sampling period is only during 2019-2020?
The sampling period on the three sites are 2018-2020 (Table 1). It is the reason why we present air masses back trajectories according to the period 2018-2020. We just remind the reviewer that 2018 database is not used for annual precipitation characterization (because of a non-satisfactory sampling representativity) (Table 1)) but monthly 2018 VWM concentrations are calculated (according good quality indicators of quarterly PCL%, see table 1). To be clear: we only remove the first 2018 trimester (Jan-Feb-March 2018) for Abidjan and Korhogo but we calculate monthly VWM for the others 2018 months.

Lines 547-551: some references are missing for this assertion: Done (now line 518)
Ref Laouali et al., 2012

Lines 572-579: Is the Abidjan site located upwind or downwind of the urban area? Since the authors have shown a high contribution from marine air masses, could this sulphate be related to other transport other than road-transport ones (i.e. shipping emissions)?

This sulphate could effectively be originated from shipping emissions since in the calculations of the marine contribution in Abidjan, we have 7% of marine fraction. This marine part of the sulphate could come from the shipping emissions but we have no reliable method to confirm it.

Table 4. Please increase the space between the lines. Hard to read. Done
Table 4 became Table 2

Line 605: are highly correlated with r value of (r=0.79), (r=0.70), (r=0.73). All the r are redundant. Done (now line 482)

Lines 671-675 and elsewhere: when comparing the values with literature, the authors should include references, even if they are detailed in table 4. Done

Line 741: Which is the Benin site? Done (now line 604)
We precise Djougou, in Benin

Figure 7: Please correct the square brackets Done (now line 626)
Now Figure 7 is Figure 6

Figure 8: quality needs to be improved. Please explain the X in kgX.ha-1. yr-1 (also present in other parts of the manuscript). What is the t.carb variable? Done
Figure 8 is moved to the supplementary materials, it became Figure S2. tcarb is now defined in Figure S2 as followed: the total carbonates species, calculated from this equation tcarb = $10^{(pH-5,505)}$ (Kulshrestha et al., 2003).

Lines 795-797: should be moved to the section when describing figure 4. done
We consider that this part should remain in the section that deals with the acidity of rain because we address each contribution section by section so we talk about organic acids in the section dedicated to the acid contribution. That's why we consider it will be appropriate to keep this part in the acid contribution.

Lines 803-805: To evaluate organic content, the authors must address the OC or DOC content. Evaluating four or five organic ions is not enough to reach any conclusion.
In this study we specifically study organic acids, not total or dissolved organic content. Of the organic acids measured in this work, formic acid, acetic acid and oxalic acid have been identified as the most common ones in both cloud and rain waters (Sun et al., 2016; Niu et al., 2018). We therefore consider, like in previous papers (e.g. Bakayoko et al., 2021, Akpo et al., 2015), that the measured organic acids are representative of organics contribution to rain water acidity.

Lines 826-830: Could the differences be related to the limited number of ions evaluated?
We don't think that the differences could be related to the number of ions that are analyzed. The number of ions evaluated in this work is not limited, as the ions measured in this work are the major ions as shown by many other authors cited in the paper. For instance, the study of Vet et al, 2014, that represents an overview paper of reference from the GAW/WMO program, presents a global assessment of rain composition and wet deposition at the global scale considering the same major ions analyzed in our study. It may still be possible, but highly unlikely, that some of the differences could come from some unacknowledged ions.

Section 3.2 is too long and repetitive. I would suggest reducing it by at least half and moving to the first part of the results in order to avoid redundancies. Done

Table A1 and Figure A2 (should be A1) must be improved. The text is very small, and the resolution is very low. Done

There are several typos, caps lock in the middle of sentences, double spaces, etc. English need to be revised and improved.
We agree and tried to carefully review the text to correct all the typos. We also revised the English.

Table 2 has been removed from the main text and placed in Appendix (referenced Table A2).

**PCA analysis results** We performed a principal component analysis (PCA) over the variables describing rain composition for each studied location. The results are displayed on Fig. 1 and analyzed below.

[Figure]

*Figure 1:* PCA results (*Abidjan-Lamto-Korhogo*)

In Abidjan, we note that the main factor of the PCA is composed of the variables $NO_3^-$, $K^+$, $Mg^{2+}$, $Ca^{2+}$, $Cl^-$, $SO_4^{2-}$, $Na^+$, $C_2O_4^{2-}$, $NH_4^+$ which are positively grouped around the axis representing the main dimension 1 with 44.42 % of the explanatory information of the existing relations between the variables. This strong proximity between these different variables in Abidjan could be explained by the fact that these ions coming from different sources participate in chemical reaction processes that are intimately linked such as acidification and neutralization. If we couple these results with the correlation factors, we see that the variables composing the first factor all have relatively good correlations with correlations distinguishing sources such as the marine source $Na^+$ and $Cl^-$ (0.94), the terrigenous source $Mg^{2+}$ and $Ca^{2+}$ (0.81), the anthropogenic source $SO_4^{2-}$ and $NO_3^-$ (0.74) and a possible fourth source (biomass fires or charcoal) with the strong correlations between $K^+$ and $Cl^-$ (0.75)

The second factor of the principal component analysis in Abidjan is mainly composed of the $HCOO^-$, $CH_3COO^-$ and $C_2H_5COO^-$ ions, with 15.48% of the explanatory information of the existing relationships between the variables. When we couple these results with the correlation factors, we see that the $HCOO^-$ and $CH_3COO^-$ ions have a very good correlation (0.75) while the $C_2H_5COO^-$ ion shows no good relationship with the other two ions. This pattern could mean a common origin of $HCOO^-$ and $CH_3COO^-$ ions which are the most abundant low molecular weight organic carboxylic acids in the global troposphere. They can either be emitted from direct sources such as vehicle exhaust emissions, biomass burning, biofuels, fossil fuels, and vegetation, or they can be formed in the atmosphere through photochemical reactions (Cruz et al., 2018). In the urban context of Abidjan, the most likely source could be vehicle emissions.

Lamto shows a rather different principal component analysis with the main factor of the PCA composed of the variables grouped and positively correlated on the axis representing dimension 1 with 52.15 % of the explanatory information of the existing relationships between the variables which are the following ions: $Ca^{2+}$, $NO_3^-$, $C_2O_4^{2-}$, $NH_4^+$, $CH_3COO^-$, $SO_4^{2-}$, $HCOO^-$. This strong correlation between these ions coming from different sources could be explained by the same mechanism that prevails at the Abidjan site, i.e. the interactions related to the acidification and neutralization process in rain water. Indeed, the very good correlation factors between the different ions ($Ca^{2+}$, $NH_4^+$) ; ($Ca^{2+}$, $SO_4^{2-}$) ; ($Ca^{2+}$, $HCOO^-$) ; ($Ca^{2+}$, $HCOO^-$) ; ($Ca^{2+}$,$Mg^{2+}$) ; ($Mg^{2+}$, $SO_4^{2-}$) ; ($Mg^{2+}$, $Cl^-$) respectively 0.87, 0.76, 0.82, 0.86, 0.92, 0.88, 0.71 are an indicator of the neutralization capacity of cations on acidic compounds and according to (Lu et al. , 2011) are probably the result of the reaction of alkaline species rich in $Ca^{2+}$ and $Mg^{2+}$ with sulfuric, nitric, hydrochloric and organic acids.

The second factor (15.00%) is composed of $Na^+$, $Cl^-$ and $K^+$ ions which are also contributing to dimension 1. Indeed, the strong correlation factors between $Na^+$ and $Cl^-$ (0.82) and $K^+$ and $Cl^-$ (0.71) confirm that these ions may come from the marine and biomass burning sources respectively (Lara et al., 2001).

Korhogo exhibits a principal component analysis on rainfall composition that is almost similar to that of Abidjan. The main factor of the PCA is composed of $NO_3^-$, $K^+$, $Mg^{2+}$, $Ca^{2+}$, $Cl^-$, $SO_4^{2-}$, $Na^+$, $C_2O_4^{2-}$, $NH_4^+$ which are positively clustered around the axis representing the main dimension 1 with 52.54% of the explanatory information of the existing relationships between the variables. The second factor (17.30%) of the principal component analysis in Korhogo is composed of $HCOO^-$, $CH_3COO^-$ and $C_2H_5COO^-$ ions. The same interpretations as for the Abidjan site are likely valid for the Korhogo site.

In conclusion, on the three sites of the South-North transect, rainfall is strongly influenced by acidification and neutralization processes related to alkaline and acidifying species emitted by various sources (anthropogenic and natural sources). However, there are differences between the urban sites of Abidjan, Korhogo and the rural site of Lamto. At the urban sites, organic species do not participate enough in the acidification process and may not come from the same source as at the rural site of Lamto.

We performed a principal component analysis on the rainfall ion concentrations at each site for our study and came to the conclusion that this analysis showed the same findings as the correlation matrix. Therefore, since we need to reduce the manuscript size, we chose not to add the principal component analysis to the article.

---

## Author Comment (AC2)

**REVIEWER COMMENTS TO AUTHOR**

The authors thank the reviewers for their useful comments. We recognize that some manuscript sections were too long and repetitive. We improved the manuscript by reducing and synthesizing all the sections. In addition, we added a co-author (C. Mouchel-Vallon) to this paper who helped significantly in this review process, especially in the acid contribution section, and in the global rewriting of the paper.

Answers and explanations are given here, accompanied by detailed descriptions of the modifications brought to the manuscript. As suggested by Reviewer #1, we moved some content to newly created appendix and supplementary material.

Additionally, we revised the abstract and the conclusion to better reflect the content of the paper.

**Reviewer 2**

General comments

**General comment 1**: The Tcarb is not described in the paper and the results are not showed.

**Response to General comment 1**

We have defined and added it to the table of acronyms. Results of tcarb calculations are in the table A1 in appendices and in Figure S2. tcarb is now defined as followed: the total carbonates species, calculated from this equation tcarb = $10^{(pH-5,505)}$ (Kulshrestha et al., 2003).

**General comment 2**: The marine and the crustal contribution are calculated assuming that 100% of $Cl^-$ and $Ca^{2+}$ are marine and crustal. But anthropogenic sources of $Cl^-$ (biomass combustion KCl) and $Ca^{2+}$ (cement production and road resuspension) are discussed in the paper. Isn't it possible to have an estimation of their anthropogenic contributions?

**Response to General comment 2**

We assumed that $Cl^-$ is almost 100 % from marine source because The production of sea-salt aerosol by wind stress at the ocean surface dominates the global emission flux of particulate C1 and of total inorganic CI; on a global scale, other sources are relatively insignificant (Keene et al., 1999).But it is true that a some part could come from others sources such biomass combustion KCl, but we assume here that it can be neglected.

For $Ca^{2+}$, we assume that it is 100% of terrigenous origin but we should make the difference between the natural origin and the anthropogenic source. Calcium naturally comes from the erosion of rocks rich in calcium. It is therefore a natural element constitutive of eroded materials (sand) or of quarry deposits. As sand and material coming from quarry deposits are widely used in construction and the production of cement, we can assume that a significant fraction of $Ca^{2+}$ could come from anthropogenic activities. So, when we mention that calcium is 100% from crustal origin, as we refer to the sources here, we are talking about any process, either natural or anthropogenic, leading to the emission of $Ca^{2+}$. However, we cannot at this stage distinguish natural vs. anthropogenic crustal $Ca^{2+}$. To do it, we need additional gas phase and particulate matter measurements and also transport chemistry model data as well as detailed emission inventories for our three sites. for the moment, we do not have these data that is the reason we have just specified that terrigenous contribution in urban areas in this study is a mixture of

anthropogenic sources and crustal sources. Whereas this terrigenous contribution in rural areas is a mixture of biomass burning sources and crustal sources.

*General comment 3:* Data values equal to 0.000 should be replaced by the limit of detection. The authors should revise carefully the values given in the text (e.g. there are very often errors in the values).

**Response to General comment 3**

We have fixed this problem in revising all values given in the text and replaced 0.00 by 'limit of detection (LOD)' in table A1 in appendices.

*General comment 4*: The authors should rewrite several parts of the paper, and the conclusion should be clearer on the main findings (e.g. no conclusion was given on the dry/wet seasons or the impact of airmasses on rainwater composition)

*Response to General comment 3*

We have carefully revised all the document and rewrite some parts, especially the conclusion and the abstract in order to better highlight main findings.

**Minor comments review#2**

Line 22: what is a climate zone?

Climate zone has been removed from the abstract to focus more on the geographical location of sites. The notion of climatic zones is now detailed in section 3 (results) to present the climatology of each site. Now line 258-264

Lines 91-92: Arrange the references in chronological order by date of publication Done Now line 95-96

Lines 112-114: repetition of sentences in lines 105-107

See the same comment from reviewer 1 Lines 112-114: Redundant, already said in lines 105-107. The sentence has been removed and some elements complete now the sentence in lines 105-107.

Figure 1: unreadable text for the map a) – missing word c)

Figure 1 has been modified done

Lines 139- 143: to be shorten

In the aim to globally shorten the paper, we decided that information's given lines 139-143 were not absolutely necessary and these lines have been removed.

Line 150-151: what is the contribution of the different emission sources?

In the article of (Keita et al., 2018), percentages of contributions are given for regional (West Africa) and particularly in Côte d' Ivoire. For example:  The regional contribution from each sector is presented in Fig. 4. The residential sector contributes to more than 50 % of BC emissions in Eastern and Western Africa. Waste burning is the second largest source of BC emissions in Eastern and Western Africa. For NOx, the traffic sector is the largest contributor in Western with 30 % of the total emissions, respectively (Fig. 4b). In Côte d'Ivoire, the residential sector is the most important (58 %), followed by waste burning (26 % with 16 % residentially and 10 % in dump locations), traffic (9 %), other sectors (5 %), industry (1 %) and energy (0.1 %), and I added reference of (Keita et 2018) in this line which move to line XXX

Line 161: delete "ha" after "2000" done, move to line 154

Line 164: add "before "Taxi-motos" done, move to line 156

Line 168: please define the V Baoule, done (now line 159)
We wrote: "…at the tip of the "V Baoule" which represents an ecological zone of transition between the forest and the savanna".

Line 176: delete "is" before "influenced", Done

Line 181: remove capital letter "These" Done
We have deleted this part of document

Line 187-192: Is the rain gauge collocated with the rainwater sampler at Abidjan?
Yes, the rain gauge is colocated with the rain sampler at Abidjan (added in the text)

2.2 climatology: technical details are missing for the rainfall data at the Korhogo site
We move results of this section in the result section 3. The description of meteorological parameters is kept in Sect. 2.2. We specified in this section that rainfall data for Abidjan and Korhogo were provided by SODEXAM, the company that collects weather and climate data for the entire country.

Figure 2: for a better representation should add -1 and 1 in the fig (in dotted line for eg.) Done

Line 224: replace June by May Done

Line 225: replace September by October Done

Line 243 – 251: revise the maximum and minimum temperature (e.g., Line 243 30.83, max in fig 2 seems to be around 29.5 in March). Done
Indeed, this was a typo, in addition this part was moved to the results and discussion part we have revised it while trying to reduce this part now entitled "Climatology sites" in order to meet one of the concerns which is to reduce the paper. This part in move now in line 305 - 309:

"Annual mean air temperature and relative humidity over the study period in the transect Abidjan -Lamto-Korhogo are respectively 27.30 °C ± 1.10 and 80 % ± 3.89, 28.98 °C ± 1.10 and 77,05 % ± 5.53 and 27.00 °C ± 0.08 and 60 % ± 0.81 (Figure 2a,2b,2c). Temperature ranged from 25.3±0.20°C in august to   28.65±1.85 °C in march, from 27.15°C ± 0.21 in august to 31.48±0.25 °C in february and from 25.2 ± 0.30 °C in August to (29.50 ± 0.26 °C) in April respectively in Abidjan, Lamto and Korhogo. (Figure 2a,2b,2c)."

Line 258 – 268: revise the number of years of surplus and deficit (e.g., Line 259 from 1980 to 2020 we have 41 years but not 20+18). Done we have revised the numbers and this part are moved in line 310 -330:

"According to the classification of (Sarr, 2009), in Abidjan, over the 41 years, 20 years are in excess while 18 years are in deficit and 3 years have values close to the mean rainfall (1522± 518 mm). The studied period (AII) index analysis shows that 2020 (AII = + 0.14) is a

moderately wet year while 2018 (AII = -0.08) and 2019 (AII =-0.31) are moderately dry years. In Lamto, over the 23 years, 8 years are in excess while 13 years are in deficit and 2 years have values close to the mean rainfall (1229 ± 165 mm). The studied period (AII) index analysis revealed that 2019 (AII =+1.7) can be considered as a strongly wet period while 2018 (AII =-0.8) and 2020 (AII =-0.8) are classified as moderately dry years. Finally, in Korhogo, over the 31 years, 12 years are in excess whereas 18 years are in deficit compared to the mean rainfall of 1187 mm and 1 years have a value close to the mean rainfall (1187 mm± 179). The studied period (AII) index analysis revealed that all three years are in deficit, with (AII) index values of -0.14, -0.15, -0.58 respectively and can be considered as moderately dry periods."

Table 1: replace "inter annual" by "interannual" Done

Line 284: "INDAAF" already define line 79 Done

Line 289: add the collection surface of the sampler Done
For the needs of reduction of the document, this part has been deleted, the reader being invited to read the article references which describe fully the procedures of collection of rain samples. But the collection surface is 225 cm$^2$

Line 292: what is the time delay before analysis? The time delay before analysis is 6 months maximum, all the procedures are respected so that the samples reach LAERO in Toulouse in the best conditions (added in the text).

Line 366: define "VCD" Done see answer giving to the rewiever 1

Line 389: replace "NSS" by "NSSF" Done see answer giving to the rewiever 1

Line 410: remove capital letter "The" Done

Line 418: why several arrival altitudes are used in the study? Generally, people are tending to use 500m or 0.5 PBL
We have used several altitudes because the West African monsoon system is responsible of most of the rains in West Africa. So, to better understand the atmospheric dynamics controlling this phenomenon we choose to scrutiny several arrival altitudes atmosphere dynamics below 3500 meters (11,500 feet) because it is at his height that most of rainy clouds are located.

Figure 3: poor quality Done, Figure 3 has been moved in supplementary materials, it became Figure S1 and quality has been improved

Line 495: 2017 is not included in the study period
Yes, it is true, chemical composition of rain is studied over 2018-2020. However, to describe seasonal variations and define season periods for each site, we have included 2017 because of seasons overlapping over two consecutive years. In order to include the whole dry season that starts in December 2017 and ends in February 2018 we have included 2017 to generate seasonal back-trajectories

Line 557: assuming that Ca$^{2+}$ is 100% crustal,
SO$_4^{2-}$, Mg$^{2+}$, K$^+$ and Cl$^-$ ratios and crustal EF are calculated assuming that Ca$^{2+}$ is 100% crustal (Table A3)

We assume that most of the calcium would come from terrigenous sources, however, depending on the context of the study area, whether urban or rural, there are nuances. in urban areas, calcium can have several origins, while in rural areas the main source remains the terrigenous source.

Table 4: results at Abidjan, Lamto and Korhogo are not in accordance with Table A1 (eg. pH in Abidjan is 5.78 in table 4, 5.76 in table A1)
Yes, that was a typo, it has been corrected, Table 4 became Table 2.

Line 607: NSSF not shown in table 4 but in table 3
This was a typo since we have modified the paper in order to shorten it, we have moved the right table in appendices section. The new legend of this table is "Table A3"

Lines 615-637: Add references to table 3, figure 3 and so on Done

Lines 631-633: the authors indicate that air masses are "heavily loaded with dust particles" but back trajectories calculated from the HYSPLIT model just help you to understand the transport pathways of air masses, there are no evidences on the time and space aerosol load.
Indeed, there is no evidence of aerosol load. However, because the transition period (March-April) between the dry season and the rainy season, during which the back-trajectories showed that air masses originated globally from the Sahara and recorded highest calcium peaks, it is a period recognized in West Africa monsoon system as the period when the ITCZ is at its southernmost position, favoring the harmattan dusty air masses which are generally loaded with calcium. Therefore, we have based our conclusion on this deduction. several authors have pointed this out in their works (Laouali et al., 2012; Marticorena et al., 2010).

Line 731: replace 0.53 by 0.58 find in Line 580 Done

Line 754: the authors indicate that "the values of nitrogen wet deposition remain lower than the critical load, estimated to be 10 kg N ha$^{-1}$ yr$^{-1}$". The concept of critical load defines the exposure of an ecosystem to acidification, eutrophication. Please precise if you assess the critical load exceedances for acidity or for eutrophication, and detail also how the critical load was calculated.
We calculate these critical loads for the eutrophication phenomenon, the calculation method has been presented in these studies (Bobbink et al., 2010; Vries et al., 2010)

Line 767-783: Do you have studied the origins of the air masses for the more acidic rains?
No, we have not done that we should consider this aspect.

Line 791 and 792: Table 5 doesn't exist Done this was a typo.

Line 795: the organic contribution to acidity is defined using sulphate and nitrate. But line 789 sulfuric and nitric acids are parts of mineral acids. It is confusing.
That is true, it was a typo and we delete sulfate and nitrate. Move to line 656

Line 803: add "organic" in "the high acidity contribution" Done, move to line 664

Figure 8: what it "Ht" in the legend?
Ht represents rain depth (mm) Figure 8 became Figure S2

Figure 9: add blue dots for site location Done Figure 9 became Figure 7 we add a legend

Line 1005: suppress "in Korhogo", this part belongs to the section2.3 which has been deleted in order to reduce the paper.

Line 1052: replace 5.54 by 5.57 (see table A1) Done, move to line 717

table A4: add INDAAF in the list of acronyms Done, move to Appendices